JCB Journal of Cell Biology

# N-cadherin dynamically regulates pediatric glioma cell migration in complex environments

Dayoung Kim[1], James M. Olson[2,3], and Jonathan A. Cooper[1]

**Pediatric high-grade gliomas are highly invasive and essentially incurable. Glioma cells migrate between neurons and glia, along axon tracts, and through extracellular matrix surrounding blood vessels and underlying the pia. Mechanisms that allow adaptation to such complex environments are poorly understood. N-cadherin is highly expressed in pediatric gliomas and associated with shorter survival. We found that intercellular homotypic N-cadherin interactions differentially regulate glioma migration according to the microenvironment, stimulating migration on cultured neurons or astrocytes but inhibiting invasion into reconstituted or astrocyte-deposited extracellular matrix. N-cadherin localizes to filamentous connections between migrating leader cells but to epithelial-like junctions between followers. Leader cells have more surface and recycling N-cadherin, increased YAP1/TAZ signaling, and increased proliferation relative to followers. YAP1/TAZ signaling is dynamically regulated as leaders and followers change position, leading to altered N-cadherin levels and organization. Together, the results suggest that pediatric glioma cells adapt to different microenvironments by regulating N-cadherin dynamics and cell–cell contacts.**

## Introduction

Pediatric high-grade gliomas (PHGGs) and diffuse midline gliomas (DMGs) resemble adult glioblastomas in being aggressively invasive and essentially incurable but differ in having fewer oncogenic mutations and arising in the growing brains of children (Azzarelli et al., 2018; Jones et al., 2012). Invasion occurs along a variety of routes, with cells moving individually, as collective strands, or as multicellular networks between neuronal and glial cell bodies, along white matter tracts, or through extracellular matrix (ECM)-rich spaces under the leptomeninges and around blood vessels (Alieva et al., 2019; Cuddapah et al., 2014; Gritsenko et al., 2012; Kluiver et al., 2020; Neve et al., 2017; Osswald et al., 2015). The varied invasion routes and dynamic morphologies of migrating glioma cells, including branching migration, locomotion, and translocation, resemble those of migrating neural progenitors and immature neurons during brain development, suggesting that similar molecular mechanisms may be at play (Beadle et al., 2008; Cuddapah et al., 2014; Venkataramani et al., 2022; Zarco et al., 2019). Molecular and phenotypic profiling reveals multiple tumor subclones with distinct invasive capacities that cooperate in vitro (Vinci et al., 2018), but the molecular and cellular mechanisms contributing to invasion in different microenvironments are poorly understood.

The ability of glioma cells to migrate either between other cells or through ECM implies differential regulation of cell–cell and cell–matrix adhesion (Wu et al., 2021). Intercellular adhesion between glioma cells may allow them to coordinate their movements during collective invasion, while glioma cell adhesion to normal cells may aid invasion into the brain parenchyma (Collins and Nelson, 2015; Friedl and Mayor, 2017; Gritsenko et al., 2020; Ng et al., 2012). The complex invasion routes and plasticity between individual and collective migration imply that glioma cells can adapt to making contacts with ECM, normal cells, or other tumor cells, but the specific molecules involved are unknown.

The cell–cell adhesion molecule N-cadherin (*CDH2*, N-cad) is upregulated in ∼60–80% of adult glioblastomas and is associated with increased mortality (Gravendeel et al., 2009; Noh et al., 2017). High N-cad expression is also associated with poor overall survival in all types of pediatric brain cancer (Fig. S1 A). During normal development, N-cad is critical both to maintain cell–cell adhesion in the neuroepithelium and to stimulate single-cell migration of neural progenitors and neural crest cells (Ganzler-Odenthal and Redies, 1998; Hatta and Takeichi, 1986; Jossin and Cooper, 2011; Kadowaki et al., 2007; Kawauchi et al., 2010; Kon et al., 2019; Theveneau et al., 2010; Vassilev et al., 2017; Xu et al., 2001). Thus N-cad has the potential to regulate single-cell or collective migration in gliomas.

At the molecular level, N-cad is a transmembrane receptor. Extracellular homophilic interactions between N-cad molecules

[1]Basic Sciences Division, Fred Hutchinson Cancer Center, Seattle, WA, USA; [2]Clinical Division, Fred Hutchinson Cancer Center, Seattle, WA, USA; [3]Ben Towne Center for Childhood Cancer Research, Seattle Children's Research Institute, Seattle, WA, USA.

Correspondence to Jonathan A. Cooper: jcooper@fredhutch.org.

on adjacent cells can bind cells together or repel them through contact inhibition of locomotion (Scarpa et al., 2015). Surface N-cad also functions cell-autonomously to regulate other transmembrane receptors such as the fibroblast growth factor receptor and integrins (Collins and Nelson, 2015; Kon et al., 2019; Mui et al., 2016; Nguyen et al., 2019). Inside the cell, the N-cad cytoplasmic domain interacts with p120-catenin, β-catenin, and α-catenin. The catenins link N-cad to the actin cytoskeleton and regulate N-cad traffic through the endosomal recycling pathway. Unbound cadherins are rapidly endocytosed and recycled (Le et al., 1999; Reynolds and Roczniak-Ferguson, 2004). N-cad recycling is important during astrocyte migration to replace surface N-cad at the cell front (Peglion et al., 2014). Catenins also regulate cell migration through Rho GTPases and Wnt signaling (Harris and Peifer, 2005; Reynolds and Roczniak-Ferguson, 2004; Vassilev et al., 2017).

In this study, we report that N-cad inhibits PHGG cell migration on ECM but stimulates migration on neurons or astrocytes, consistent with glioma–glioma cell interactions slowing and glioma–neural cell interactions speeding migration. In addition, N-cad is localized and regulated differently in PHGG leader and follower cells during migration. N-cad localizes to filamentous connections that join leader cells to each other and to adjacent neurons. In contrast, follower cells are tightly packed, forming linear, epithelial-like N-cad junctions with their neighbors. The leader cells have higher cell proliferation and nuclear localization of the transcription coactivator Yes-associated protein 1 (YAP1) as well as increased surface N-cad when compared with follower cells. YAP1 nuclear localization is determined by cell density and its activation increases N-cad surface level and recycling. Thus, YAP1 activation at the migration front increases N-cad surface levels, promoting collective migration and invasion in neural environments.

## Results

### N-cad inhibits PHGG migration on ECM but stimulates migration on neural cells

We investigated the migration properties of five PHGG and DMG cell lines by transferring tumor spheroids to different environments designed to mimic the ECM or neural cells (neuron or astrocyte) through which invasion occurs in vivo. All cell lines migrated collectively in all environments, with very few single cells. Even as cells spread away from the spheroid, they remained connected, suggesting an important role for cell–cell adhesion molecules. Curiously, migration in ECM environments did not correlate with migration over neurons. For example, one MYC-N amplified PHGG line, PBT-05, migrated faster on neurons but slower on laminin or reconstituted basement membrane (Matrigel) than the transcriptionally similar PBT-04 line (Fig. S1, B–D). We hypothesized that this paradoxical migration behavior might result from the altered expression of cell adhesion molecules. N-cad is the most highly expressed cell–cell adhesion molecule at the RNA level in most PHGG lines (Brabetz et al., 2018) (Fig. S1 E and Table S1) and was readily detected at the protein level in the three DMG and two PHGG lines that we

analyzed (Fig. S1 F). Interestingly, N-cad is more highly expressed in PBT-05 than PBT-04 cells, correlating with migration on neurons but inversely correlating with migration on ECM.

To directly test whether N-cad differentially regulates migration in different environments, we established PBT-05 lines expressing control or N-cad shRNA and characterized them by Western blot and RNASeq. N-cad shRNA inhibited N-cad expression at the protein (Fig. S1 G) and RNA level (sevenfold decrease, CDH2, Fig. S1 H). Expression of other cadherins was unaffected except for CDH3 (2.8-fold increase, Fig. S1 H and Table S2). Importantly, integrin expression was also unaffected. N-cad depletion did not affect the spheroid formation or DNA synthesis (Fig. S1, I–K) but stimulated migration on laminin and invasion into Matrigel (Fig. 1, A–D, Video 1; and Fig. S1, N and O). Migration on laminin was also stimulated by transient knockdown of N-cad with siRNA (Fig. S1, L and M). Thus, N-cad inhibits migration on ECM. In contrast, migration on cultured cerebellar neurons or mouse astrocytes was inhibited by N-cad depletion (Fig. 1, E–H; and Fig. S1 P). Astrocyte cultures secrete a 3D ECM scaffold mimicking the brain parenchyma (Gritsenko and Friedl, 2018). To test whether N-cad depletion inhibits glioma migration on astrocyte cultures through glioma cell interactions with astrocyte cells or ECM, we used decellularized astrocyte cultures. N-cad depletion stimulated migration on astrocyte-derived, decellularized ECM, suggesting that N-cad stimulates glioma migration on astrocyte cells but not ECM (Fig. 1, I and J; and Fig. S1 Q). These results show that N-cad differentially regulates migration according to the environment, increasing migration on neural cells (neurons or glia) but inhibiting migration on ECM (laminin, Matrigel, or astrocyte-derived ECM).

N-cad also regulated the balance between collective and individual cell migration. In ECM or neural cell environments, N-cad depletion increased the number of PBT-05 cells that migrate individually away from the collective and decreased the number of contacts between cells at the migration front (Fig. 2, A–C; and Fig. S1, N and O). The remaining connections at the migration front were elongated and resembled the filamentous junctions or tumor microtubes that connect cell networks in adult glioblastoma (Fig. 2 B, arrowhead) (Osswald et al., 2015; Venkataramani et al., 2022). Quantification of individual cell movements revealed that N-cad depletion decreased glioma cell directionality on neurons but increased directionality on laminin, without significantly affecting the average migration speed in either environment (Fig. 2, D–I). These findings suggest that N-cad promotes collective relative to single-cell migration in multiple environments but has environment-specific effects on cell migration directionality and overall speed, stimulating directionality and overall distance migrated in neural and astrocyte environments while inhibiting directionality and overall migration and invasion into ECM.

### N-cad stimulates and inhibits migration through intercellular homotypic interaction

At the molecular level, N-cad can regulate cell migration through multiple mechanisms, including intercellular N-cad homotypic interactions, cell-autonomous regulation of receptor kinases,

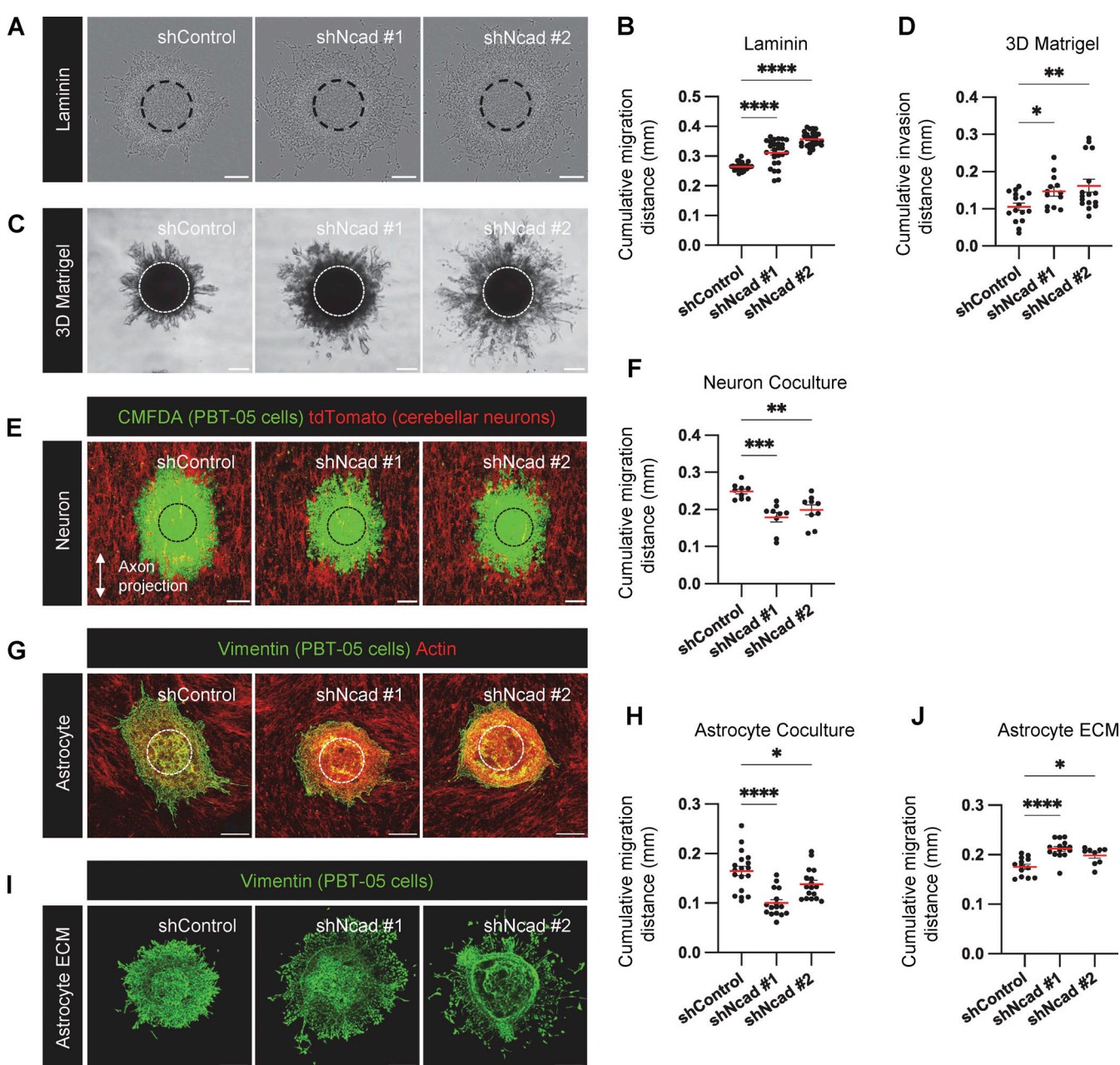

Figure 1. **N-cad inhibits PHGG migration on ECM but stimulates migration on neurons and astrocytes. (A and B)** Representative images and quantification of cumulative migration distance for control or N-cad shRNA PBT-05 spheroids plated on laminin for 24 h **(C and D)** Representative images and quantification of invasion in 3D Matrigel for 96 h. **(E and F)** Images and quantification of migration on neurons for 72 h. PBT-05 spheroids were labeled with cell-permeable Green CMFDA fluorescent dye before transferring to tdTomato-expressing cerebellar neurons that had differentiated along aligned nanofibers. **(G and H)** Images and quantification of migration on mouse astrocytes for 48 h. PBT-05 cells were detected with human-specific anti-vimentin antibodies. Actin was detected with phalloidin. **(I and J)** Images and quantification of migration on astrocyte-derived decellularized ECM for 24 h. (A, C, E, and G) Dashed circles represent spheroid at 0 h. Scale bars, 200 µm. (B, D, F, H, and J) Error bars indicate mean ± SEM. Ordinary one-way ANOVA Holm-Šídák's multiple comparisons tests. *P < 0.05, **P < 0.01, ***P < 0.001, ****P < 0.0001. **(B)** N = 27–29 spheroids, three experiments. **(D)** N = 13–16 spheroids, four experiments. **(F)** N = 9 spheroids, three experiments. **(H)** N = 16–18 spheroids, three experiments. **(J)** N = 9–15 spheroids, two experiments.

and intracellular catenin signaling (Khalil and de Rooij, 2019; Kon et al., 2019; Theveneau et al., 2010; Vassilev et al., 2017). We wondered whether intercellular N-cad homotypic interactions might explain its conflicting roles in migration in different environments. For example, neurons or glia ahead of the migration front might increase directionality and stimulate overall migration, while other glioma cells behind the migration front

might decrease directionality and inhibit overall migration. This model predicts that N-cad makes glioma–neuron junctions as well as glioma–glioma junctions. Indeed, we detected N-cad in filamentous contacts between migrating PBT-05 cells and cerebellar neurons (Fig. 3 A). To test whether intercellular N-cad contacts were important for migration, we expressed an Ncad$^{W161A}$ mutant that is incapable of forming strand-swapped

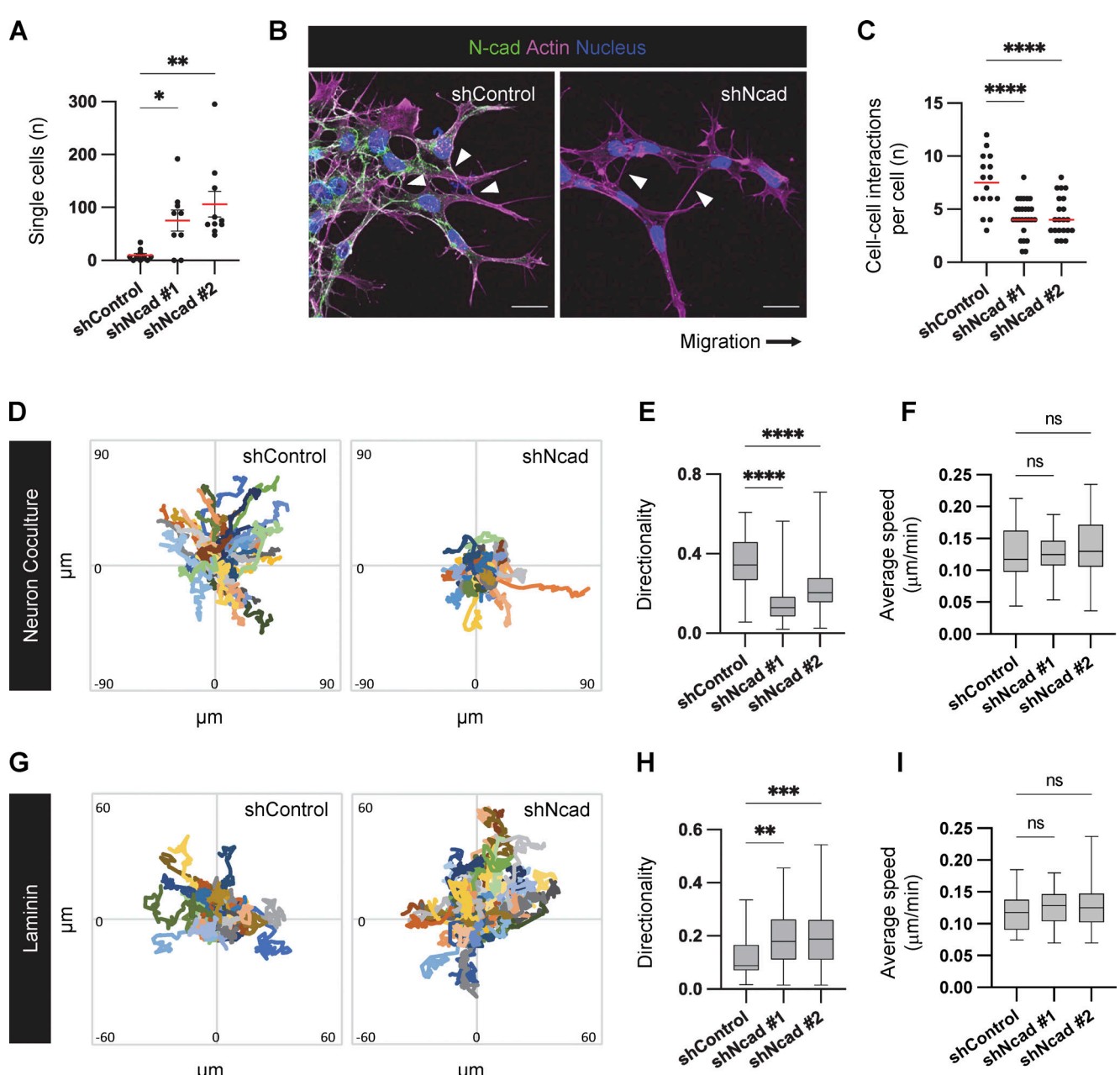

Figure 2. **N-cad regulates PHGG cell cohesion during migration. (A)** Number of single cells migrating ahead of the migration front after 24 h migration on laminin. **(B and C)** Images and quantification of cell-cell connections at the migration front. Scale bar, 20 μm. Error bars show mean ± SEM. **(D–I)** Cell tracking. Dendra2-histone H2B–expressing control or N-cad shRNA cells migrating on neurons or laminin for 16 h. Cell tracks (D and G) and box and whisker plots (E, F, H, and I) indicating median, quartiles, and range. Ordinary one-way ANOVA Holm-Šídák's multiple comparisons tests. ns, not significant, *P < 0.05, **P < 0.01, ***P < 0.001, ****P < 0.0001. **(A)** N = 12 spheroids. **(C)** N = 16–26 spheroids, three experiments. **(E and F)** N = 42–102 cells, three spheroids. **(H and I)** N = 49–70 cells, three spheroids.

homotypic dimers (Shapiro et al., 1995; Tamura et al., 1998). Ncad^W161A and Ncad^WT were tagged with mCherry and expressed at a similar level (Fig. 3 B). Ncad^W161A expression mimicked N-cad depletion, inhibiting cell migration on neurons but stimulating migration on laminin (Fig. 3, C and D; and Fig. S2, A and B), consistent with intercellular homotypic interactions stimulating and inhibiting migration in different environments. To directly test whether N-cad in the environment stimulates migration, we assayed migration on a surface coated

with the N-cad extracellular domain (ECD). Glioma cells migrated rapidly on N-cad ECD, dependent on endogenous N-cad (Fig. 3 E). Moreover, glioma cells migrated poorly over N-cad-depleted astrocytes (Fig. 3, F and G; and Fig. S2 C). Together, these findings suggest that intercellular N-cad homotypic interactions with neural cells ahead of the migration front speed overall migration, while interactions with other glioma cells behind the migration front slow overall migration (Fig. 3 H).

**Figure 3. Intercellular N-cad homotypic interactions slow PHGG migration on ECM and speed migration on neurons and astrocytes. (A)** N-cad in intercellular contacts between migrating PBT-05 cells, labeled with Green CMFDA, and cerebellar neurons expressing tdTomato. Migration for 72 h before fixation and staining with N-cad antibody. *xy* images are maximum intensity projections from multiple *z*-planes. Insets show enlarged *xy, xz,* and *yz* images. Arrowheads indicate N-cad at cell–cell contacts. Scale bars, 20 and 5 µm (inset). **(B)** Western blot analysis for Ncad^WT-mCherry or Ncad^W161A-mCherry expression in PBT-05 cells. β-Tubulin is shown as a loading control. **(C)** Cumulative migration distances on neurons for 72 h. **(D)** Cumulative migration distance on laminin for 24 h. **(E)** N-cad in the environment stimulates PHGG migration. Migration on poly-D-lysine or N-cad extracellular domain (ECD)-Fc surfaces for 24 h. **(F and G)** Astrocyte N-cad stimulates PHGG migration. **(F)** Western blot shows N-cad siRNA depletes N-cad from mouse astrocytes. β-Tubulin is shown as a loading control. **(G)** Cumulative migration distances on control or N-cad-depleted astrocytes for 48 h. **(H)** Model diagram: migration on ECM is slowed by N-cad-mediated glioma-glioma interactions but migration on neurons or astrocytes is accelerated by N-cad-mediated glioma binding to surrounding neural cells. **(C, D, and G)** Unpaired *t* test. **(E)** Ordinary one-way ANOVA Šídák's multiple comparisons test. Error bars show mean ± SEM. *P < 0.05, ***P < 0.001, ****P < 0.0001. **(C)** N = 8–11 spheroids, three experiments. **(D)** N = 14–16 spheroids, three experiments. **(E)** N = 5 spheroids, two experiments. **(G)** N = 13–16 spheroids, three experiments. Source data are available for this figure: SourceData F3.

### Catenins regulate N-cad surface levels to stimulate or inhibit migration

Cadherins and catenins reciprocally coregulate: cadherins localize catenins to the membrane while catenins regulate cadherin surface retention and internalization, cytoskeletal linkage, and signaling (Harris and Tepass, 2010; Katsuno-Kambe and Yap, 2020; Nanes et al., 2012; Reynolds and Roczniak-Ferguson, 2004). We tested whether catenins regulate N-cad expression or surface levels in PHGG and their effect on migration. Depletion of p120-, β-, or α-catenin individually had little effect on the steady-state protein levels of other catenins or of N-cad (Fig. 4, A and B). However, in each case, the steady-state level of N-cad on

the cell surface was reduced (Fig. 4, C and D). Remarkably, catenin depletion increased cell migration on laminin and inhibited migration on astrocytes, in parallel with the N-cad surface level (Fig. 4, E and F; and Fig. S2, D and E). These findings suggest that catenins stimulate or inhibit glioma cell migration in different environments primarily by regulating N-cad surface levels.

To test whether N-cad regulates catenin protein levels or localization, we investigated catenin expression and localization in control and N-cad-depleted cells. Western blotting showed that N-cad depletion significantly decreased the level of β- but not α- or p120-catenin and induced a mobility shift in p120-catenin, possibly indicating a change in phosphorylation state (Reynolds and Roczniak-Ferguson, 2004) (Fig. S2, F and G). We used immunofluorescence to localize N-cad and catenins in mixed cultures of control and N-cad-depleted cells. In control cells, N-cad and all three catenins colocalized on the plasma membrane (Fig. S2 H, arrowheads), with additional localization of N-cad, β-, and α- but not p120-catenin in perinuclear vesicles (Fig. S2 H, arrows). In cells lacking N-cad (dashed regions in Fig. S2 H), both the surface and vesicular localization of β- and α-catenin was decreased, while p120-catenin decreased at the surface and increased in the cytoplasm consistent with release during endocytosis (Davis et al., 2003). Overall, these findings are consistent with N-cad regulating catenin localization, while catenins regulate N-cad surface levels with corresponding effects on cell migration.

## Altered cell–cell contacts and increased N-cad levels, endocytosis, and recycling at the migration front

Studies of cancer invasion suggest that "leader cells" at the migration front and "follower cells" behind the front are functionally distinct, and that cooperation between leaders and followers increases overall migration (Konen et al., 2017; Zhang et al., 2019). We noted that N-cad localizes to distinct cell–cell contacts between PHGG leaders and followers: radially oriented, neurite-like, filamentous cell–cell contacts between leaders (white arrowheads, Fig. 5 A, r2, see also Fig. 2 B), and circumferential, epithelial-like, contacts between follower cells (yellow arrowheads, Fig. 5 A, r1). Filamentous N-cad junctions also link leader cells to neurons (Fig. 3 A). These structures appear to be similar to filamentous junctions or tumor microtubes that form between infiltrating adult glioblastoma cells (Gritsenko et al., 2020; Osswald et al., 2015). We also noted that leading cells contain more N-cad perinuclear vesicles compared with follower cells (white arrow, Fig. 5 A, r2), even though the number of transferrin receptor-containing vesicles is the same (Fig. 5 B). Intracellular N-cad vesicles colocalized extensively with early endosomes (Rab5), at intermediate levels with recycling endosomes (Rab11), and poorly with lysosomes (LAMP1) or the Golgi (GM130), suggesting that N-cad endocytosis and recycling may be increased in leader relative to follower cells, with possible functional significance for migration (Fig. 5, C and D).

To test whether N-cad endocytosis is regulated differently in leaders and followers, we used an antibody uptake assay. Migrating cells were cooled to 4°C and incubated with an antibody to the N-cad extracellular domain. Unbound antibody was removed and cells were transferred to 37°C at different times. Cells were fixed and surface and internalized N-cad antibodies were detected using two different fluorochrome-conjugated secondary antibodies (Fig. S3 A). Initially, most N-cad antibody was on the surface, but over time it progressively entered perinuclear endosomes that stained for early endosome markers EEA1 and Rab5 and recycling markers Rab4 and Rab11 (Fig. 5 E; and Fig. S3, B and C). Quantification showed that leader cells internalized more N-cad antibody than followers (Fig. 5 F). With longer incubation, more N-cad antibody returned to the surface of leaders than followers implying more recycling (Fig. S3 D). Indeed, a modified assay confirmed that more N-cad antibody recycled to the surface in a leader than in follower cells (Fig. 5 G; and Fig. S3, E and F). Together, these results show that leader cells internalize and recycle more N-cad antibody than followers. However, we noted that leader cells also bound more N-cad antibody initially (Fig. 5 E). This implies increased surface, internal and total N-cad protein levels. Increased N-cad protein was unexpected since we show below that N-cad RNA is equal in leader and follower cells.

We took two approaches to test whether N-cad protein levels are actually higher in leader than follower cells. First, migrating cells were fixed, permeabilized, stained with N-cad antibody, and imaged, and fluorescence intensity was integrated across multiple z sections. Total N-cad intensity was significantly higher in leaders than followers (Fig. 5 H). Second, we measured surface N-cad levels in leader and follower cells by flow cytometry (Fig. S3, G–K). PBT-05 cells were transduced to express Dendra2, a photoconvertible fluorescent protein, fused to histone H2B. Cell nuclei of leader and follower cells were irradiated to photoconvert histone H2B-Dendra2 from green to red (leaders) or yellow (followers) (Fig. S3, G and H). Cells were then released from the plate without the use of proteases (to avoid cleaving N-cad), labeled with N-cad antibody, and the N-cad intensity in leader and follower cells was measured by flow cytometry (Fig. S3, I–K). N-cad surface levels were significantly higher in leader than follower cells.

Together, the evidence suggests that distinct N-cad cell–cell junctions form between migrating PHGG leader and follower cells, and that these distinct junctions correlate with increased total, surface, and endosomal levels of N-cad protein in leader cells. However, it is unclear whether the altered N-cad levels, subcellular distribution, and cell junctions are a cause or consequence of cell position in the migratory stream and the types of cell contacts formed.

## Altered proliferation, YAP1/TAZ signaling, and gene expression of leader cells

In light of the altered N-cad protein levels and distribution in PHGG leaders and followers, we performed additional phenotypic characterization. Leader cells from epithelial-derived cancers such as breast and lung carcinomas express different genes and proliferate more slowly than followers (Konen et al., 2017; Zhang et al., 2019). To test whether PHGG leaders proliferate more slowly than followers, we pulse-labeled migrating cells with EdU (Buck et al., 2008). EdU incorporation was significantly higher in leader than follower cells whether migrating on laminin or neurons or invading Matrigel (Fig. 6, A and B).

**Figure 4. Catenins regulate N-cad levels and cell migration. (A and B)** Western blot analysis of control, p120-catenin (p120-ctn), β-catenin (β-ctn), and α-catenin (α-ctn) in PBT-05 cells expressing various shRNAs. Protein levels were normalized to β-tubulin as a loading control. Two-way ANOVA uncorrected Fisher's LSD test. $N$ = 3 Western blots. **(C and D)** Representative histograms and quantification of N-cad surface levels in control and catenin-depleted cells measured by flow cytometry. Ordinary one-way ANOVA with Dunnett's multiple comparisons test. $N$ = 3 experiments. **(E and F)** Cumulative migration distances on (E) astrocytes for 48 h and (F) laminin for 24 h. Ordinary one-way ANOVA Dunnett's multiple comparisons test. $N$ = 7–11 spheroids, two experiments. Error bars show mean ± SEM. *$P < 0.05$, **$P < 0.01$, ***$P < 0.001$, ****$P < 0.0001$. Source data are available for this figure: SourceData F4.

Thus, unlike the situation in breast and lung cancer, PHGG leader cells are more, not less, proliferative than followers.

YAP1 and TAZ (*WWTR1*) are related transcriptional co-activators that are regulated by cell density and stimulate cell proliferation, plasticity, and migration (Panciera et al., 2017; Totaro et al., 2018; Zanconato et al., 2019). YAP1 and TAZ are nuclear in subconfluent cells but excluded from the nucleus in dense cultures, mediated in part by signals from cadherins (Kim

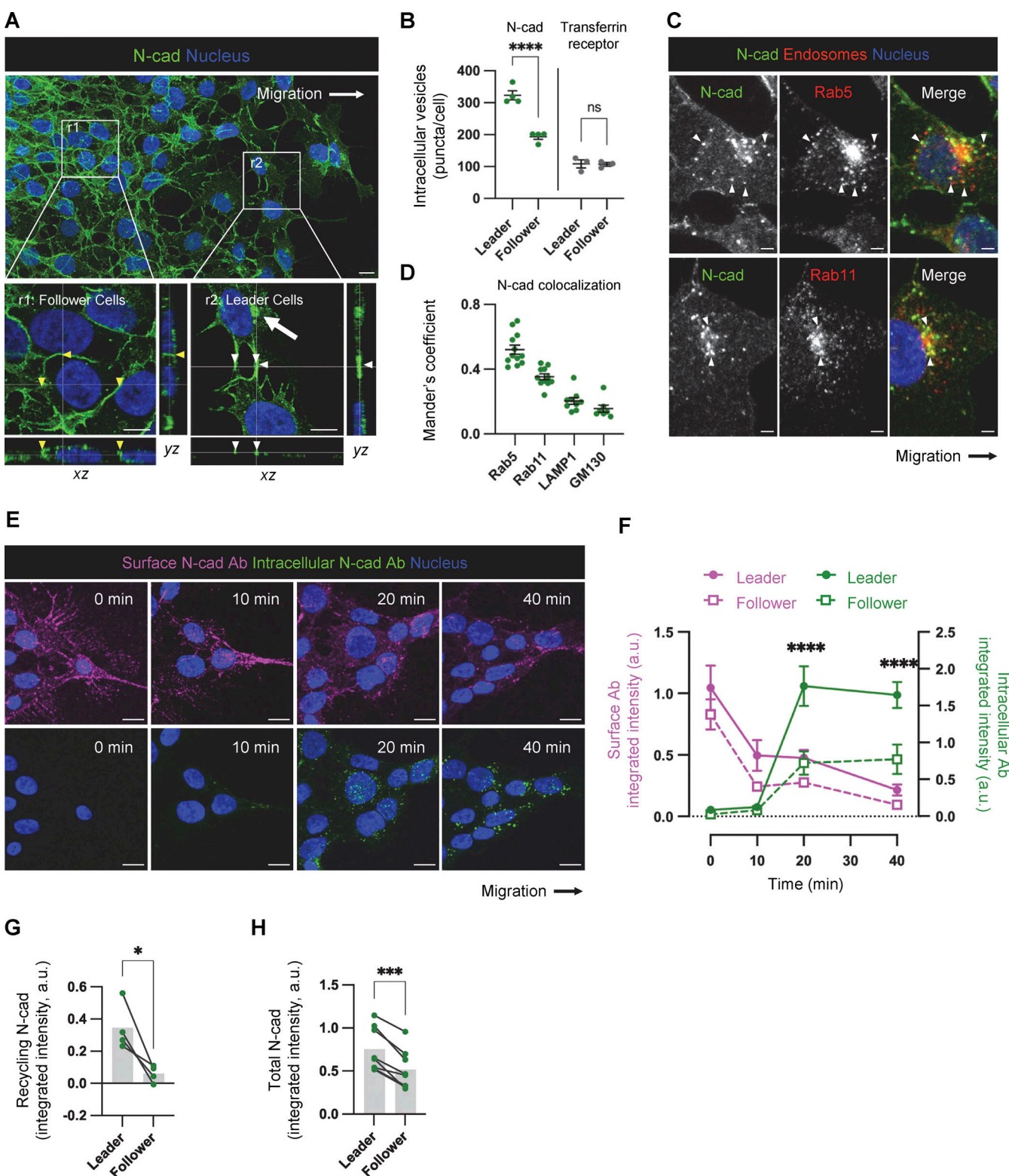

Figure 5. **Distinct N-cad localization and dynamics in leader and follower cells. (A)** Localization of N-cad in PBT-05 cells migrating on laminin. Maximum intensity projections. Insets show enlarged orthogonal views of follower (r1) and leader (r2) cells. Arrowheads indicate cell-cell junctions and arrows indicate vesicles. Scale bars, 10 or 7 μm (inset). **(B)** Intracellular N-cad or transferrin receptor-positive vesicles per cell were counted in leader and follower cells. *N* = 3 experiments. Unpaired *t* test. ***P < 0.0001. ns, not significant. **(C)** Representative images showing colocalization between N-cad and endosomal markers. Arrowheads indicate N-cad-positive endocytic vesicles colocalized with Rab5 or Rab11. Scale bars, 10 or 2 μm (inset). **(D)** Mander's colocalization coefficients. *N* = 7–12 cells, 2–3 spheroids for each marker protein analyzed. **(E and F)** N-cad antibody uptake assay. Surface N-cad was labeled with N-cad ECD antibodies on ice before warming for various times, fixing, and staining for surface and intracellular antibody as described in Fig. S3 A. **(E)** Representative images. Scale bars, 10 μm. **(F)** Quantification. a.u., arbitrary units. *N* = 7–24 cells, 2–4 spheroids, 2 experiments. Two-way ANOVA Tukey's multiple comparisons test between

leader and follower cells at the same time point. ****P < 0.0001. **(G)** Recycling of intracellular N-cad Abs to the surface in leader and follower cells, measured as described in Fig. S3 E. N = 4 experiments each representing the average of 10 cells in each of 3–5 spheroids. Paired t test. *P < 0.05. **(H)** Quantification of total N-cad fluorescence intensity in leader and follower cells. Migrating cells were fixed, permeabilized, and stained with N-cad Ab. Fluorescence intensity was integrated for multiple leader and follower cells across the z stack. Paired t test. ***P < 0.001. N = 8 spheroids, five experiments. Error bars show mean ± SEM.

et al., 2011; Panciera et al., 2017). PHGG leader cells are less crowded and proliferate faster than followers, suggesting that YAP1/TAZ signaling may be activated. Indeed, YAP1 was nuclear in a higher proportion of leader than follower PBT-05 cells regardless of whether the cells were migrating in Matrigel, laminin, or neurons (Fig. 6, C and D). Nuclear localization of YAP1 was also higher in leader than follower cells in DMG lines PBT-24 and PBT-29 (Fig. S4, A and B), suggesting that YAP1 nuclear localization may be a common feature of leader cells during pediatric glioma migration.

We tested using RNA sequencing to compare gene expression in leader and follower cells that we isolated by photoconversion and flow cytometry, as described above (Fig. S3, G–I). Only 44 gene transcripts increased and 36 decreased >2^0.5-fold in leaders relative to followers out of 19,729 genes that were quantified (Benjamini-Hochberg adjusted P value <0.05, four independent experiments) (Fig. 6 E and Table S3). Expression of N-cad, catenins, other junctional regulators, and YAP1 and TAZ were unchanged (Fig. S4 C). However, the expression of YAP-response genes, CTGF, CYR61, ACTA2, GADD45A and CDKN1A, and wound-healing genes TNFRSF12A, CD9, TPM1, ANXA1 and ANXA5, was higher in leader than follower cells (Fig. 6 E and Table S3). Gene ontology analysis suggested a modest increase in wound healing and stress gene signatures (Fig. S4 D). This suggests that YAP1-signaling and wound-healing gene expression is increased in leader cells and may contribute to their faster migration and higher proliferation.

**Leader and follower cells are not predetermined**
Leader cells may be genetically or epigenetically predetermined and selected to lead invasion based on, for example, increased expression of promigratory genes. Alternatively, cell phenotypes may be plastic, with leader cell properties induced by the unique environment at the migration front (Khalil and de Rooij, 2019; Konen et al., 2017; Zhang et al., 2019). To better understand whether the PHGG leader phenotype is predetermined or plastic, we photoconverted histone H2B-Dendra2-expressing leader cells and monitored their positions as migration continued (Fig. 6, F and G; and Video 2). Regardless of migration conditions—neurons, laminin, or Matrigel—40–60% of leader cells were overtaken by cells from behind during the time it took to move 35–40 μm (Fig. 6 H). Furthermore, YAP1 was nuclear in cells that had recently transitioned to become leaders and cytoplasmic in cells that had recently become followers (Fig. 6 I). These findings show that leader and follower cells change position during migration and invasion, suggesting they are not predetermined. Moreover, YAP1 signaling and gene expression (Fig. S4, C and D) change as leaders and followers switch positions.

**YAP1 regulation by cell density independent of N-cad**
We tested whether reduced cell density or reduced N-cad engagement increases YAP1 nuclear translocation in leader cells.

First, we imaged YAP1 in migrating N-cad-depleted cells (Fig. 7 A). Leader nuclei contained more YAP1 than followers regardless of N-cad expression level. This suggests that N-cad does not directly regulate YAP1 nuclear localization. Second, we tried to equalize leader and follower exposure to environmental N-cad by observing cells migrating over N-cad ECD (Fig. 7, B and C). YAP1 was still largely nuclear in leader cells and cytoplasmic in followers. This result aligns with the nuclear localization of YAP1 in leader cells migrating on neurons (Fig. 6, C and D). Third, to test the role of cell crowding, control or N-cad-depleted cells were plated on a micropatterned coverslip to generate cell clusters of different sizes. The proportion of cells with nuclear YAP1 decreased as cluster size increased, consistent with larger clusters having more center cells and fewer edge cells, and hence more cell–cell contact, regardless of N-cad status (Fig. 7, D and E). Additionally, N-cad antibody uptake decreased as nuclear YAP1 decreased in large clusters, showing that there is a correlation between YAP1 and N-cad internalization (Fig. 7 F). This suggests that YAP1 is excluded from the nucleus in follower cells by cell crowding, independent of N-cad, and translocates to the nucleus of leader cells, even if N-cad is present in the surrounding environment because cells are less crowded.

**YAP1/TAZ signaling regulates cell migration and N-cad protein levels**
YAP1/TAZ signaling stimulates cell migration and invasion in other systems (Panciera et al., 2017; Totaro et al., 2018; Zanconato et al., 2019). The increased YAP1 nuclear localization and increased expression of YAP-response genes in leader cells may stimulate their migration. We tested whether YAP1/TAZ signaling regulates PHGG cell migration using YAP1 and TAZ siRNA. Overall migration was inhibited by a single knockdown of either YAP1 or TAZ and further inhibited by a knockdown of YAP1 and TAZ together (Fig. S5, A and B). YAP1/TAZ depletion also inhibited PBT-05 migration in basal media where potential confounding effects of YAP1/TAZ signaling on cell proliferation are minimized (Fig. S5 B). This suggests that the increased YAP1/TAZ signaling in leader cells may promote overall migration.

YAP1/TAZ signaling regulates cell–cell junctions and VE-cadherin turnover during vascular development (Kim et al., 2017; Neto et al., 2018). Therefore, we tested whether YAP1/TAZ signaling regulates N-cad levels or trafficking in PHGG cells. N-cad total and surface intensities were decreased in YAP1/TAZ-depleted cells (Fig. 8, A–F). Moreover, YAP1/TAZ depletion inhibited surface N-cad antibody binding in leader cells (where YAP1 is active) but not follower cells (where YAP1 is inactive) (Fig. 8, G and H), resulting in decreased N-cad antibody internalization (Fig. S5, C and D). As a control, transferrin surface level and uptake were independent of YAP1/TAZ expression (Fig. 8 F; and Fig. S5, E and F). Together, these findings suggest a

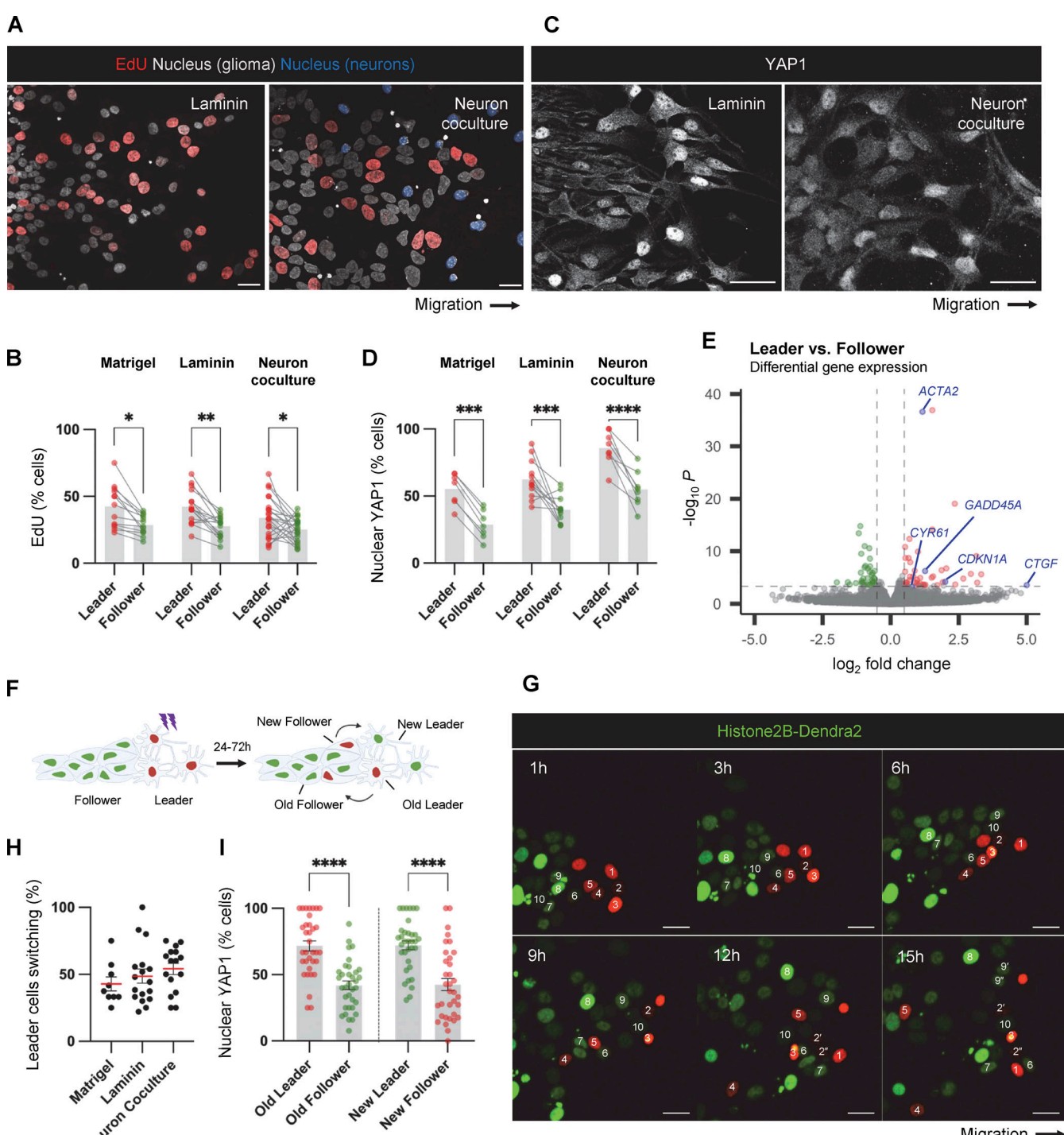

Figure 6.   **Increased DNA synthesis and YAP1/TAZ signaling in leader cells. (A and C)** EdU or YAP1 in leader and follower cells in spheroid cell migration on laminin for 24 h or neurons for 72 h. Scale bars, 20 µm. **(B and D)** Quantification of the percentage of EdU or nuclear YAP1-positive leader and follower cells. PBT-05 spheroids invaded or migrated in Matrigel for 96 h, laminin for 24 h and neurons for 72 h. **(B)** N = 13–23 spheroids, three to four experiments. **(D)** N = 7–12 spheroids, three experiments. **(E)** RNA sequencing data of leader and follower cells. Each point in the Volcano plots represents a differentially expressed gene from four biological replicates. Genes with at least $\log_2$ fold change >0.5 and FDR < 0.05 were colored. Red represents upregulated genes and green represents downregulated genes in leader cells compared with follower cells. Blue represents the YAP1-response genes. **(F)** Schematic diagram of photo-conversion of histone H2B-Dendra2 expressing leader and follower cells. **(G)** Representative images from a time-lapse movie of histone H2B-Dendra2 spheroid migration on laminin. Scale bars, 20 µm. **(H)** The percentage of leader cells switching positions with follower cells in Matrigel (48 hr after photoconversion, N = 9 spheroids), laminin (16 hr after photoconversion, 17 spheroids), and neurons (16 h after photoconversion, 16 spheroids). **(I)** Migrating histone H2B-Dendra2 expressing PBT-05 cells were fixed and permeabilized 24 h after photoconversion and processed for YAP1 immunofluorescence. The percent of old and new leaders and followers with nuclear YAP1 was calculated. N = 34 spheroids, three experiments. **(B and D)** Two-way ANOVA Šídák's multiple comparisons test. **(I)** Ordinary one-way ANOVA Tukey's multiple comparisons test. Error bars indicate mean ± SEM. *P < 0.05, **P < 0.001, ***P < 0.001, ****P < 0.0001.

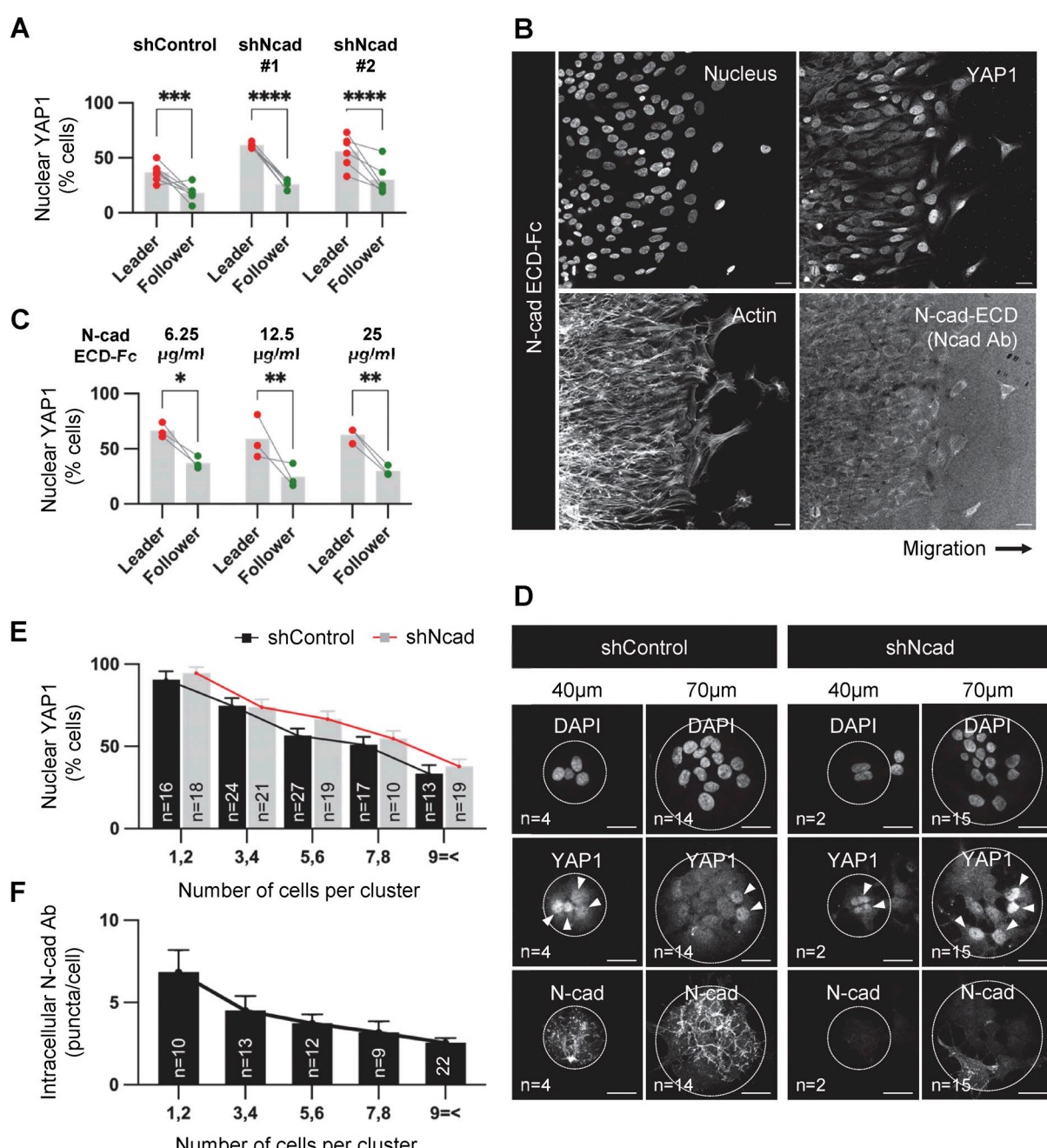

**Figure 7. YAP1 regulation by cell density independent of N-cad. (A)** Effect of N-cad depletion on YAP1 localization in migrating PBT-05 cells. **(B and C)** Effect of extracellular N-cad on YAP1 localization. Representative images of cells migrating on 12.5 µg/ml N-cad ECD-Fc coated surface for 24 h and quantification of percent nuclear YAP1 after migration on 6.25, 12.5, or 25 µg/ml N-cad ECD-Fc. **(D)** Representative images of control and N-cad shRNA cells on 40 or 70 µm laminin-coated discs for 48 h before fixation and staining for YAP1 and N-cad. The number of cells per cluster is noted in the lower left. Arrowheads indicate cells with nuclear YAP1. Scale bar, 20 µm. **(E)** Quantification of percent cells with nuclear YAP1 in different size clusters of control and N-cad shRNA cells. **(F)** N-cad antibody uptake by cells in different size clusters. **(A and C)** Two-way ANOVA Šídák's multiple comparisons test. Error bars show mean ± SEM. *$P < 0.05$, **$P < 0.001$, ***$P < 0.001$, ****$P < 0.0001$. **(A)** $N = 6$–7 spheroids, three experiments. **(B)** $N = 3$ spheroids in each condition. **(E and F)** The number of clusters quantified is noted on the graph.

model in which decreased crowding of leader cells leads to N-cad-independent nuclear translocation of YAP1 and expression of migration genes. Elevated nuclear YAP1 also increases N-cad surface levels and recycling in leader cells, which promotes migration in neural environments but inhibits migration on ECM (Fig. 8 I).

## Discussion

Our results suggest that N-cad differentially regulates glioma cell migration in neural cells or ECM environments in vitro, with higher N-cad levels increasing migration on neurons and astrocytes and lower levels increasing migration in ECM. Thus, the level of N-cad might determine the choice of migration

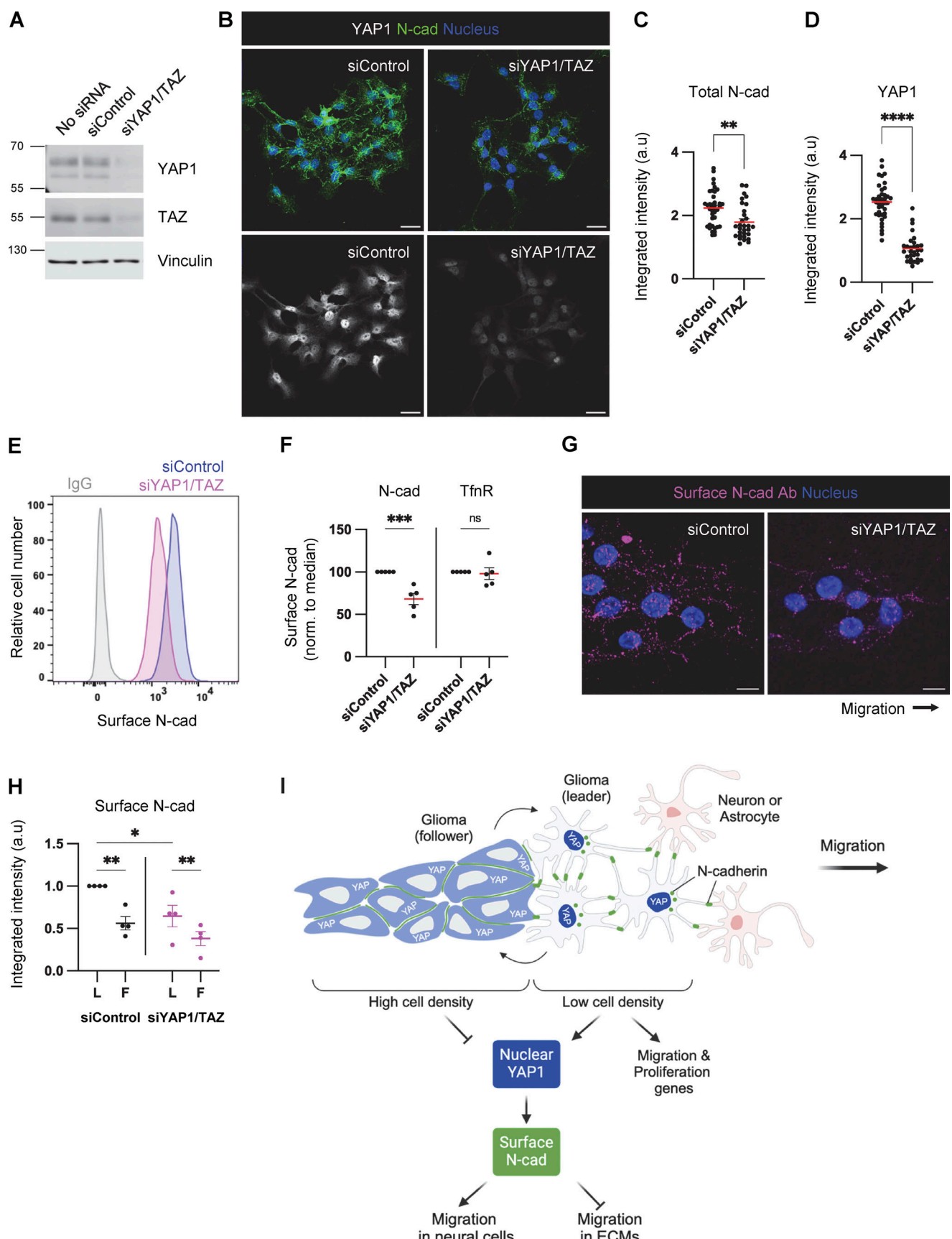

Figure 8. **YAP1/TAZ regulates N-cad surface level during PHGG migration. (A)** Western blot analysis of control or YAP1/TAZ siRNA-treated cells. Vinculin is shown as a loading control. **(B–D)** Representative images and quantification of total N-cad in low-density cultures of control and YAP1/TAZ-depleted cells.

Maximum intensity projections. Scale bars, 20 μm. a.u., arbitrary units. **(E and F)** Representative histograms and quantification of N-cad surface levels in low-density cultures of control and YAP1/TAZ-depleted cells measured by flow cytometry. **(G and H)** Representative images and quantification of surface N-cad ECD antibody binding to migrating leader (L) and follower (F) cells. Scale bars, 10 μm. **(I)** Model diagram: YAP1 activation regulated by cell density affects N-cad surface expression and stimulates migration in neurons and glia but inhibits migration in ECM. **(C and D)** Unpaired $t$ test. **(F)** Two-way ANOVA Šídák's multiple comparisons test. **(H)** Two-way ANOVA Tukey's multiple comparisons test. Error bars show mean ± SEM. *$P < 0.05$, **$P < 0.01$, ***$P < 0.001$, ****$P < 0.0001$. ns, not significant. **(C and D)** $N = 30$ cells. **(F)** $N = 5$ experiments. **(H)** $N = 4$ experiments. Source data are available for this figure: SourceData F8.

routes in vivo, switching between invading the gray matter and axon tracts as opposed to the ECM around blood vessels and under the pia. We also found that migrating leader and follower cells are phenotypically different. Leader cells are connected by N-cad-dependent filamentous junctions, have high levels of surface and recycling N-cad, and increased YAP1/TAZ signaling and proliferation compared with followers. These phenotypes of leader and follower cells are not stable, and leaders and followers exchange during migration or invasion. Thus, pediatric glioma cells adapt to their surroundings to optimize migration.

Both the pro- and anti-migratory effects of N-cad on cell migration depended on homotypic N-cad binding to N-cad in the environment or other cells. In addition, the pro- and anti-migratory effects of α-, β-, and p120-catenins mimicked those of N-cad, as expected if their main role is to stabilize N-cad on the surface to increase homotypic binding. A simple hypothesis is that leader cell migration is stimulated by N-cad engagement on the leading edge and slowed by N-cad engagement on the trailing edge (Fig. 3 H). Leading edge stimulation predominates in neural environments, but trailing edge inhibition predominates in ECM. This could be explained simply by physical force vectors, but signaling mechanisms may also play a role. For example, N-cad engagement can locally inhibit focal adhesions, inhibit Rac, and activate Rho to repress protrusion and induce cell polarization and contact-inhibition of locomotion (Camand et al., 2012; Jülich et al., 2015; Ouyang et al., 2013; Scarpa et al., 2015; Theveneau et al., 2010). However, crosstalk between N-cad and integrins is unlikely to be relevant in our system because we did not detect classic focal adhesions in either control or N-cad-deficient glioma cells migrating on laminin. Surface N-cad could alternatively regulate signaling receptors on the same cell, such as stabilizing the fibroblast growth factor receptor as occurs during cortical neuron migration (Kon et al., 2019). Catenins may also regulate directional migration from intracellular vesicles (Vassilev et al., 2017). In our system, however, surface N-cad appears to both stimulate and inhibit migration through homotypic binding or signaling interactions between cells.

N-cad surface levels and recycling were higher in leader than follower cells. N-cad recycling is needed for migration in many systems, including cortical neuron migration during development (Kawauchi et al., 2010), Xenopus neural crest (Kuriyama et al., 2014), and various cancer cells (Gritsenko et al., 2020; Sandilands et al., 2023; Wint et al., 2023). N-cad recycling may help in the disassembly and reassembly of different types of cell–cell junctions, allowing the migrating cells to rearrange and cooperate for overall migration. Collective migration and metastasis are more efficient for cancer cells with weak cell–cell adhesions (partial EMT) than when cell–cell adhesions are

completely disrupted (full EMT) (Saxena et al., 2020; Te Boekhorst and Friedl, 2016; Wu et al., 2021). Thus N-cad endocytosis and recycling may allow migrating glioma cells to reorganize filamentous connections between leader cells into tighter junctions between followers.

While leader and follower cells are predetermined in breast and lung cancer (Konen et al., 2017; Zhang et al., 2019), migrating glioma leader and follower cells switched positions, even when confined in ECM. Such switching has been observed in various biological processes, including embryonic development and cancer invasion (Cai et al., 2014; Jakobsson et al., 2010; Khalil and de Rooij, 2019; Zhang et al., 2019). In this leader–follower interchange, cells maintain cell–cell contact but continually rearrange and form different types of adhesions. For example, endothelial tip cells have serrated VE-cad junctions, high VE-cad turnover, and low cell–cell cohesion, and switch positions with follower cells that have straight, continuous VE-cad junctions, low turnover, and high cell–cell cohesion (Bentley et al., 2014). VE-cad turnover is regulated by YAP1 during vascular development (Neto et al., 2018). Similarly, the distinct N-cad distribution we observe between leader and follower glioma cells could facilitate the leader–follower interchange. Moreover, YAP1 translocates to the nucleus when cells reach the migration front, independent of N-cad but dependent on cell density. YAP1 in turn regulates N-cad level and trafficking, implying that YAP1 might contribute to the leader–follower interchange and phenotypic plasticity.

The mechanisms by which YAP1 regulates N-cad levels and trafficking remain to be explored. YAP1 is widely expressed in human brain tumors and is strongly associated with poor survival (Orr et al., 2011). Leader cells expressed higher levels of YAP1-response and wound-healing gene transcripts, but transcript levels of N-cad and proteins known to regulate cadherin traffic, such as p120-catenin (Davis et al., 2003), Rab5/11 (Kawauchi et al., 2010), and Rac1 (Akhtar and Hotchin, 2001), were similar. Therefore, N-cad is likely regulated at the level of protein synthesis or turnover. More endosomal N-cad recycled to the surface of the leader than follower cells, implying that follower cells might divert more N-cad for lysosomal degradation, but our attempts to interfere with N-cad endocytosis or degradation specifically were unsuccessful. Further understanding of the mechanism and function of N-cad recycling for glioma cell migration will require cargo-specific ways to selectively regulate endocytosis and recycling.

Cancer cells exhibit remarkable plasticity in cytoskeletal and adhesion properties that enable adaptation to varied microenvironments encountered during invasion and metastasis (Te Boekhorst and Friedl, 2016; Wu et al., 2021). This plasticity is particularly evident in adult and pediatric gliomas, which

rapidly spread by invading through different brain regions (Alieva et al., 2019; Cuddapah et al., 2014; Gritsenko et al., 2012; Kluiver et al., 2020; Neve et al., 2017; Osswald et al., 2015). Our results suggest that N-cad mediates the switch between collective or single-cell migration and allows pediatric glioma cells to adapt to their environment. YAP1/TAZ signaling is dynamically regulated as leaders and followers change position, leading to altered N-cadherin levels and organization. YAP1/TAZ inhibition may result in reduced migratory capacity without substantially reducing tumor bulk, culminating in disease control but not eradication.

## Materials and methods

### Cells
Human PHGG cell lines PBT-05 and PBT-04 cells and human DMG cell lines PBT-22, PBT-24, and PBT-29 were a kind gift from James Olson (Seattle Children's Research Institute, Seattle, WA, USA). PHGG and DMG cells were maintained in glioma growth medium (Human NeuroCult NS-A Proliferation kit, 05751; Stemcell Technologies) supplemented with 20 ng/ml human epidermal growth factor (PHG0311; Invitrogen), 20 ng/ml human basic fibroblast growth factor (78003; Stemcell Technologies), 2 µg/ml heparin (07980; Stemcell Technologies), and penicillin–streptomycin (15140-122; Gibco). Plates were coated with 20 µg/ml laminin (L2020; Sigma-Aldrich or 3446-005-01; Trevigen) for all lines except PBT-04, which were grown on uncoated plates. Cells were detached with Accutase (A6964; Sigma-Aldrich) for passage. Primary murine astrocytes immortalized with SV40 large T-antigen and H-RasV12 were kindly provided by Amparo Acker-Palmer (Max Planck Institute for Brain Research Frankfurt, Frankfurt, Germany) (Depner et al., 2016). Murine astrocytes and HEK-293FT cells (RRID: CVCL_6911) were maintained in D10 medium (Dulbecco's modified Eagle's medium [11965-092; Gibco], 10% fetal bovine serum, 2 mM L-glutamine, 1 mM sodium pyruvate, 0.075% sodium bicarbonate, and penicillin–streptomycin).

### Antibodies
Antibodies used in this study are listed in Table 1.

### DNA constructs and lentiviral transduction
Lentiviral pLKO.1-puromycin negative control and human CDH2, CTNND1, CTNNB1, and CTNNA1 shRNA vectors were from Sigma-Aldrich (Table 2).

pLenti.CAG.H2B.Dendra2.W was from Addgene (RRID: Addgene_51005). N-cad WT or W161A mutated DNA was amplified by PCR from pCAG-Ncad WT or W161A-HA (Kon et al., 2019) and cloned into lentiviral pLenti-CAG-mCherry vector using NEBuilder HiFi DNA assembly (E2621; New England Biolabs) to generate pLenti-Ncad WT-HA-mCherry or pLenti-Ncad W161A-HA-mCherry.

To harvest lentiviral particles, lentiviral vector DNA was transfected with psPAX2 (RRID:Addgene_12260) and pMD2.G (RRID:Addgene_12259) packaging plasmids into HEK-293FT cells using Lipofectamine 2000 transfection reagent (11668019; Invitrogen) and D10 media lacking antibiotics. After 24 h, media

were changed to glioma growth medium and the virus collected for a further 40 h. Culture media were collected and filtered through a 0.45-µM syringe filter. Glioma cells in a six-well plate were incubated with 500 µl of viral supernatant and 500 µl growth media for at least 48 h before selection. pLKO.1-shRNA-puromycin transduced cells were selected with 0.5 µM puromycin for 48 h. Cells expressing H2B-Dendra2, Ncad WT-HA-mCherry, or Ncad W161A-HA-mCherry constructs were sorted on a SONY MA900 Multi-Application Cell Sorter.

### Electroporation
For siRNA electroporation, $1 \times 10^6$ cells were mixed with 5 µl of 20 µM siRNAs (GeneSolution siRNA, Qiagen; Table 3) and 100 µl of Ingenio electroporation solution (MIR50114; Mirus) and nucleofected with an Amaxa Nucleofector I (Lonza) using program C-13 for PHGGs or T-20 for astrocytes. The efficiency of downregulation was validated using Western blot.

### Western blot
Cells were lysed in ice-cold 1X Triton X-100 lysis buffer (1% Triton X-100, 150 mM NaCl, 10 mM HEPES pH 7.4, 2 mM EDTA, and 50 mM NaF) with additional protease and phosphatase inhibitors (10 µg/ml Aprotinin, 1 mM PMSF, and 1 mM sodium vanadate), and insoluble material removed at 14,000 rpm for 10 min at 4°C. Protein concentrations were equalized using a Pierce BCA protein assay kit (23225; Thermo Fisher Scientific), adjusted to 1X SDS sample buffer (50 mM Tris-Cl pH 6.8, 2% SDS, 0.1 % bromophenol blue, 10% glycerol), and heated to 95°C for 5 min. Equal amounts (generally 10–20 µg) of protein were separated by sodium dodecyl sulfate-polyacrylamide gel electrophoresis and transferred to a nitrocellulose membrane. The membrane was blocked in Intercept blocking buffer (927-60003; LI-COR Biosciences) with 5% BSA for 30 min at RT. After blocking, the membrane was probed with primary antibodies overnight at 4°C, followed by IRDye 680RD goat anti-mouse or 800CW goat anti-rabbit conjugated secondary antibodies for 1 h at RT. Fluorescent images were detected using the Odyssey Infrared Imaging System (LI-COR Biosciences). Quantification was performed using FIJI.

### Spheroid migration and invasion
Spheroids were prepared from PBT-05 and PBT-04 cells by seeding $5 \times 10^3$ cells in multiple wells of ultra-low attachment round bottom 96-well plates (7007; Corning) for 72 h. Spheroids were prepared similarly from PBT-22, PBT-24, and PBT-29 cells, except the cell number was $2 \times 10^3$, and 0.192% methylcellulose (M0512; Sigma-Aldrich) was included in the media. For migration assays, single spheroids were transferred with 20-µl tips to wells in 48-well plates (3548; Corning) containing various migration substrates and incubated in 200 µl of either glioma migration medium (Human NeuroCult NS-A Proliferation kit and 20 ng/ml bFGF) or basal medium (Human NeuroCult NS-A Basal and 20 ng/ml bFGF). Phase-contrast or fluorescence images were taken at various times. The areas occupied by the spheroid, and migrating cells were outlined and quantified in FIJI. The cumulative migration distance for each spheroid is the

**Table 1.  Antibodies used in this study**

| Antibody | Supplier | Catalog | Dilution |
|---|---|---|---|
| Mouse anti-N-cad (32) | BD Biosciences | 610920 | WB (1:1,000), IF (1:200) |
| Rabbit anti-N-cad (H-63) | Santa Cruz | SC7939 | IF (1:200) |
| Rabbit anti-N-cad | Proteintech | 220181AP | WB (1:1,000), IF (1:200) |
| Rabbit anti-N-cad | Cell signaling | 13116 | IF (1:100) |
| Mouse anti-p120-catenin | BD Biosciences | 610133 | WB (1:1,000), IF (1:200) |
| Mouse anti-β-catenin | BD Biosciences | 610153 | WB (1:1,000), IF (1:200) |
| Mouse anti-α-E-catenin | Santa Cruz | SC9988 | WB (1:1,000), IF (1:200) |
| Rabbit anti-vimentin | Abcam | ab19348 | IF (1:500) |
| Mouse anti-EEA1 | BD Biosciences | 610456 | IF (1:500) |
| Mouse anti-Rab5 | BD Biosciences | 610724 | IF (1:500) |
| Mouse anti-Rab4 | BD Biosciences | 610888 | IF (1:500) |
| Mouse anti-Rab11 | BD Biosciences | 610656 | IF (1:500) |
| Goat anti-LAMP1 | Santa Cruz | SC8098 | IF (1:500) |
| Mouse anti-GM130 | BD Biosciences | 610822 | IF (1:500) |
| Mouse anti-YAP1 | Abcam | ab56701 | WB (1:500), IF (1:200) |
| Mouse anti-TAZ | BD Biosciences | 560235 | WB (1:250) |
| Mouse anti-vinculin | Sigma-Aldrich | V9131 | WB (1:10,000) |
| Mouse anti-β-tubulin | Sigma-Aldrich | T7816 | WB (1:10,000) |
| Alexa-Fluor 488 donkey anti-rabbit | Invitrogen | A21206 | IF (1:500) |
| Alexa-Fluor 568 goat anti-rabbit | Invitrogen | A11011 | IF (1:500) |
| Alexa-Fluor 647 donkey anti-rabbit | Invitrogen | A31571 | IF (1:500) |
| Alexa-Fluor 488 goat anti-mouse | Invitrogen | A11001 | IF (1:500) |
| Alexa-Fluor 568 goat anti-mouse | Invitrogen | A11004 | IF (1:500) |
| Alexa-Fluor 647 goat anti-mouse | Invitrogen | A28181 | IF (1:500) |
| Alexa-Fluor 568 donkey anti-goat | Invitrogen | A11057 | IF (1:500) |
| IRDye 680RD goat anti-mouse | LI-COR | 926-68070 | WB (1:5,000) |
| IRDye 800CW goat anti-rabbit | LI-COR | 926-32211 | WB (1:5,000) |

average distance between the migration front and the spheroid edge, calculated using the formula:

$$\text{Cumulative migration distance} = \sqrt{End\ migration\ area/\pi} - \sqrt{Start\ migration\ area/\pi}$$

**ECM substrates**

To assay migration on ECM, spheroids were transferred to wells that had been coated with 20 µg/ml laminin (L2020; Sigma-Aldrich) or 20 µg/ml fibronectin (F1141; Sigma-Aldrich) overnight, drained, and washed with basal media. Images were taken every 15 min for 48 h on an Incucyte S3 Live-cell analysis system

**Table 2.  shRNA constructs used in this study**

| MISSION shRNAs | Catalog/Clone ID | Target sequences |
|---|---|---|
| pLKO.1-puro non-target shRNA | SHC016 | 5'-GCGCGATAGCGCTAATAATTT-3' |
| pLKO.1-puro CDH2 shRNA #1 | TRCN0000312701 | 5'-GTGCAACAGTATACGTTAATA-3' |
| pLKO.1-puro CDH2 shRNA #2 | TRCN0000053978 | 5'-CCAGTGACTATTAAGAGAAAT-3' |
| pLKO.1-puro CTNND1 shRNA | TRCN0000344830 | 5'-ACTACCCTCCTGATGGTTATA-3' |
| pLKO.1-puro CTNNB1 shRNA | TRCN0000314921 | 5'-TCTAACCTCACTTGCAATAAT-3' |
| pLKO.1-puro CTNNA1 shRNA | TRCN0000234534 | 5'-CCCTCTGTCCTCAGGTTATTA-3' |

**Table 3.  siRNAs used in this study**

| siRNAs | Catalog | Target sequences |
|---|---|---|
| Negative control siRNA | 1027310 | 5'-AATTCTCCGAACGTGTCACGT-3' |
| *CDH2* siRNA | 1027417 | 5'-CTGAGCTCAGTTACACTTGAA-3' |
| Human *YAP1* siRNA 5 | 1027416 | 5'-CAGGTGATACTATCAACCAAA-3' |
| Human *YAP1* siRNA 8 | | 5'-CCGGGATGTCTCAGGAATTGA-3' |
| Human *YAP1* siRNA 6 | | 5'-CACATCGATCAGACAACAACA-3' |
| Human *YAP1* siRNA 7 | | 5'-TTGAAGTAGTTTAGTGTTCTA-3' |
| Human *TAZ (WWTR1)* siRNA 4 | 1027416 | 5'-ACAGTAGTACCAAATGCTTTA-3' |
| Human *TAZ (WWTR1)* siRNA 3 | | 5'-AGACATGAGATCCATCACTAA-3' |
| Human *TAZ (WWTR1)* siRNA 2 | | 5'-CTGGCTGTAATCACTACCATT-3' |
| Human *TAZ (WWTR1)* siRNA 1 | | 5'-CTGCGTTCTTGTGACAGATTA-3' |

(4×/0.2; Sartorius). The area covered by migrating cells was measured with the Incucyte analysis program (Sartorius).

### Invasion

To assay invasion into Matrigel, wells were first coated with 30 µg/ml Matrigel (356231; Corning) overnight at 4°C and washed with basal media. Spheroids were added and allowed to attach for 1 hr at 37°C, then covered with 100 µl 5 mg/ml Matrigel. After an additional 1 h incubation at 37°C, 100 µl glioma migration medium was added on top of the Matrigel-embedded spheroids. Phase-contrast images were taken every 24 h on Nikon Eclipse TS100. The area covered by invading cells was measured with FIJI.

### Neurons

To assay glioma migration on neurons, granule neuron cultures were prepared from euthanized wild-type or Rosa26$^{mT/mG}$ (expressing membrane-targeted tandem dimer Tomato) postnatal day 5 mice. Cerebella were dissected in dissection solution (1X Hank's balanced salts solution [HBSS; Gibco], 2.5 mM HEPES, 35 mM glucose, 4 mM sodium bicarbonate, and 1.2 mM MgSO$_4$) and dissociated with 0.1% trypsin and 0.25 mg/ml deoxyribonuclease I (D5319; Sigma-Aldrich) in dissection solution for 10 min at 37°C. After tissue fragments had settled, cells were diluted in a dissection solution containing deoxyribonuclease I and centrifuged at 200 × $g$ for 5 min. The cell pellet was suspended in granule neuron growth medium (1X Basal medium eagle [Gibco], 10% fetal bovine serum, 1 M KCl, and 1X penicillin–streptomycin) and seeded in 0.1 mg/ml poly-D-lysine (P0899; Sigma-Aldrich) precoated plates for 20 min at 37°C to deplete glia. Unattached cells were collected and centrifuged at 200 × $g$ for 5 min, resuspended in a growth medium, and viable cells counted. 1 × 10$^6$ cells were seeded on 0.5 mg/ml poly-D-lysine-coated sterile 12-mm diameter coverslips or aligned 700-nm diameter nanofibers in 24-well plates (Z694533; Sigma-Aldrich). The growth medium was changed on days in vitro (DIV) 1 and 3, adding 10 µg/ml cytosine arabinoside (AraC, C6645; Sigma-Aldrich) to reduce the growth of non-neuronal cells. Glioma cells were seeded into 96-well ultralow attachment round-bottom plates for spheroid production on day 2,

including 2.5 µM of CytoTrace Green CMFDA (22017; AAT Bioquest) in the media to facilitate cell tracking. On the morning and afternoon of DIV5, half of the neuron growth medium was replaced with the glioma migration medium. Glioma cell spheroids were rinsed with the NeuroCult basal medium to wash out the remaining CytoTrace Green CMFDA and then transferred to the neurons together with 250 µl glioma migration medium. Images were acquired every 24 h for 3 d on a Zeiss LSM780 confocal microscope (10×/0.45 air objective). The area covered by migrating cells on neurons was measured with FIJI.

### Astrocytes

To assay migration on astrocytes, we adapted the 3D astrocyte-derived scaffold procedure described previously (Gritsenko et al., 2017). 1 × 10$^5$ immortalized murine astrocytes were cultured on 30 µg/ml Matrigel-coated 12-mm glass coverslips in D10. Astrocytes rapidly acidify glioma migration medium, so, after 2 d in D10, we added 5 mM 2-deoxy-D-glucose (D8375; Sigma-Aldrich) and 400 nM rotenone (557368; Sigma-Aldrich) to inhibit metabolism. At day 4, glioma cell spheroids were transferred on top of astrocyte scaffolds with 500 µl glioma migration medium and cultured for a further 2 d. Cells were fixed with 4% paraformaldehyde (PFA, 100503-917; VWR), for 15 min at room temperature (RT) and stained with human-specific anti-vimentin antibodies to visualize glioma cells. Images were acquired on Leica Stellaris 5 (10×/0.40 air objective). The area covered by migrating cells was quantified with FIJI.

### Astrocyte ECM

Astrocytes were cultured as described above for 4 d, then lysed and the ECM was washed as described previously (Harris et al., 2018). To confirm astrocyte removal, matrices were stained with anti-N-cad antibody, phalloidin, and DAPI. Glioma cell spheroids were allowed to migrate for 24 h before staining with anti-human vimentin antibody and quantification of the area covered.

### N-cad ECD Fc

A 96-well ELISA high-binding plate (9018; Corning) was coated with 12.5 µg/ml recombinant human N-cad Fc chimera protein

(Asp160-Ala724, 1388-NC; R&D systems) or 20 µg/ml poly-D-lysine in coating buffer (HBSS with 1 mM $CaCl_2$) overnight at 4°C. Wells were then blocked with 3% BSA in HBSS for 2 h at RT and washed three times in HBSS supplemented with 1.2 mM $CaCl_2$. Glioma spheroids were transferred to the precoated plates and incubated in glioma migration medium for 24 h at 37°C, 5% $CO_2$.

### EdU incorporation

Cells were incubated with 5 µM EdU (5-ethynyl-2′-deoxyuridine) diluted in glioma growth medium for 1 h (laminin and Matrigel) or 3 hr (neurons) at 37°C. Cells were fixed with 4% paraformaldehyde for 15 min and permeabilized with 0.5% Triton X-100 for 20 min at RT. EdU was detected with the Click-it 488 reaction solution (C10337; Invitrogen) for 30 min. Nuclei were stained with Hoechst 33342 for an additional 30 min.

### Cell tracking

Cell movements were tracked using histone H2B-Dendra2 PBT-05 cells. Spheroids were transferred to laminin or neurons for 6 or 24 h, respectively before starting imaging at 15-min intervals using an Andor Dragonfly spinning disk confocal microscope (10×/0.45 air objective; Oxford instruments) and a humidity, $CO_2$ (5%), and temperature (37°C) controlled chamber. The nuclear movement was tracked using Imaris (RRID:SCR_007370; Oxford Instruments), and the directionality and velocity of individual cells were quantified using a custom-made open-source computer program, DiPer (Gorelik and Gautreau, 2014).

### Immunofluorescence

Cells were grown or allowed to migrate on 12-mm diameter coverglasses precoated with an appropriate substrate. Cells were fixed with 4% PFA for 15 min at RT, permeabilized with 0.1% Triton X-100 in PBS for 10 min at RT, and blocked with 5% normal goat serum (005-000-121; Jackson Immunoresearch), 2% BSA in PBS for 1 h at RT. Primary antibodies were diluted in 1% BSA in PBS and incubated overnight at 4°C. Alexa Fluor-conjugated goat anti-mouse or donkey anti-rabbit IgG (H+L) secondary antibodies were incubated together with Alexa Fluor-conjugated phalloidin (A12380 or A22287; Invitrogen) for 1 h at RT. After 4′,6-diamidino-2-phenylindole (DAPI, 1:5,000) incubation for 5 min at RT, the coverglass was mounted with Pro-Long Glass Antifade Mount (P36984; Invitrogen). Confocal imaging was performed on a Zeiss LSM780 or a Leica Stellaris 5 confocal microscope (63×/1.40 oil objective).

### N-cad antibody internalization and recycling

Glioma cell spheroids were allowed to migrate on laminin-coated coverslips for 24 h. Coverslips were transferred to 4°C and incubated in a glioma migration medium containing 10 µg/ml rabbit anti-N-cad antibody (22018-1-AP; Proteintech), which recognizes the N-cad extracellular domain. After 30 min, cells were washed three times with Human Neurocult NS-A basal medium and then fed with glioma migration medium. Cells were incubated for 0, 10, 20, 40, or 60 min at 37°C and fixed with 4% PFA for 15 min at RT. For recycling, cells were allowed to internalize the antibody for 40 min at 37°C, cooled to 4°C, and the remaining cell surface antibody was blocked by incubation with 0.13 mg/ml F(ab')$_2$ fragment goat-anti rabbit IgG (111-006-045; Jackson Immunoresearch) for 30 min. The cells were then washed and incubated for an additional 20 min at 37°C to allow recycling of internalized antibody before fixation. To differentially detect surface and internalized antibody, we adopted a procedure described in Carrodus et al. (2014). Fixed cells were blocked with 5% BSA in PBS for 30 min at RT and incubated with donkey-anti-rabbit IgG (H+L) Alexa Fluor 647 (1:200) to label the cell-surface bound N-cad antibodies. After 2 h, the remaining rabbit IgG on the surface was blocked with a solution of 0.13 mg/ml AffiniPure Fab fragment goat-anti-rabbit IgG (H+L), 5% BSA in PBS, overnight at 4°C. After blocking, cells were refixed with 4% PFA for 5 min at RT, then permeabilized and blocked with 0.1% Triton X-100 and 5% BSA in PBS for 30 min at RT. Another secondary antibody, donkey-anti-rabbit IgG (H+L) Alexa Fluor 488 (1:200), was used to label internalized N-cad antibodies. Additional antibodies were included at this stage to analyze N-cad colocalization with markers of specific subcellular compartments. Confocal images were acquired on Zeiss LSM780 or Leica Stellaris 5 (63×/1.40 oil objective).

The number of cell surface or intracellular N-cad antibodies-positive puncta was measured using Imaris. The surface creation tool was used to automatically detect endosomal vesicles. The estimated XY diameter for surface detection was 0.5 µm. The background was subtracted. Colocalization of intracellular N-cad antibodies with other proteins was measured by overlapping volume between 3D surfaces of two proteins. Integrated intensities were measured using FIJI.

### Transferrin uptake

Unlabeled transferrin was removed by preincubating cells in Human NeuroCult NS-A basal medium for 30 min at 37°C. Cells were then cooled and incubated with 100 µg/ml Alexa Fluor 568-conjugated transferrin (T23365; Sigma-Aldrich) for 30 min at 4°C. Cells were washed three times with the basal medium and incubated for 40 min at 37°C before fixation and imaging.

### Photoconversion

To isolate leader and follower cells for quantification of N-cad surface levels or RNA isolation, histone H2B-Dendra2 PBT-05 cell spheroids were formed for 3 days and transferred to laminin. After 24 h, migrating leader and follower cells from 80 to 90 spheroids were selectively photoconverted for 5,000 or 1,000 ms using the 405 nm laser (FRAPPA photobleaching module, Andor iQ3 Mosaic) on an Andor Dragonfly spinning disk confocal microscope equipped with Mosaic and a humidity, $CO_2$ (5%), and temperature (37°C)-controlled chamber.

To monitor the switching of leader and follower cells during migration, histone H2B-Dendra2 PBT-05 cell spheroids were placed on laminin or neurons or embedded in Matrigel and allowed to migrate for 6, 24, or 48 h respectively. Leader cells were then photoconverted for 5,000 ms as above. Images were taken after an additional 16, 16, or 48 hr respectively, and the number of red follower cells was calculated as a percentage of total red cells.

## N-cad surface quantification by flow cytometry

To quantify surface N-cad by flow cytometry, cells were detached with 2 mM EDTA in HBSS for 10 min, resuspended in HBSS containing 0.5% BSA, and incubated for 30 min at 4°C with 10 µg/ml anti-N-cad primary antibody (22018-1-AP; Proteintech). Cells were washed twice in HBSS/0.5% BSA and incubated with Alexa Fluor 647-conjugated donkey anti-rabbit IgG (H+L) secondary antibodies (1:200) for 30 min at 4°C. Cells were analyzed by flow cytometry (BD FACSymphony A5; BD Bioscience). Data were analyzed with FlowJo (RRID:SCR_008520, v10.9.0).

## RNA sequencing

To compare RNA transcriptomes from bulk cell populations, control or N-cad shRNA cells were dissociated using Accutase at room temperature for 10 min and resuspended in HBSS. 500 cells were collected in the center of a 1.5 ml centrifuge tube containing 4.75 µl SMART-seq reaction buffer (Takara) using a BD FACSymphony S6 (BD Bioscience), avoiding cell loss on the tube walls. To compare RNA transcriptomes from leader and follower cells, ~90 spheroids of PBT-05 cells expressing histone H2B-Dendra2 cells were allowed to migrate for 24 h on laminin and photoconverted as above. After photoconversion, cells were dissociated with Accutase at room temperature, resuspended in HBSS, transferred to ice, and ~200 leader and follower cells were sorted into SMART-seq reaction buffer as above. Each experiment was performed on four different occasions.

RNA was prepared and cDNA was synthesized with the SMART-Seq v4 Ultra Low Input RNA Kit for Sequencing (Takara) and run on the Agilent Tapestation to assess the cDNA product. To construct RNA sequencing libraries, we used Illumina's Nextera XT kit to fragment the cDNA and added barcoded sequencing adapters. Differential gene expression analysis was performed with the DEseq2 (RRID:SCR_015687) for paired sample R package (Love et al., 2014). Genes with a Benjamini-Hochberg adjusted P value <0.05 were defined as differentially expressed.

## Micropattern adhesion

A micropatterned glass coverslip with disc diameters from 10 to 100 µm (http://4dcell.com) was precoated with 20 µg/ml laminin overnight at 37°C. $1 \times 10^5$ cells were resuspended in 1 ml of glioma migration medium and plated on the laminin-coated micropatterned glass coverslip for 1 h at 37°C. Unattached cells were carefully removed, and fresh glioma migration medium was added. After 48 h, cells were fixed and stained with YAP1 and N-cad antibodies. For N-cad antibody internalization, cells were allowed to internalize N-cad ECD antibodies for 40 min at 37°C before fixation and staining as described above. The number of puncta was divided by the number of nuclei in each cluster to calculate N-cad antibody internalization per cell.

### Statistical analysis and graphics

The number of replicates for each experiment is described in the figure legends. All experiments were repeated in at least three independent biological replicates, otherwise noted in the figure legend. Statistical analysis and data graphing were performed using GraphPad Prism 9 and 10 software (RRID: SCR_002798). P values were determined using unpaired $t$ test, paired $t$ test, ordinary one-way ANOVA Holm-Šídák's, Šídák's, Tukey's multiple and Dunnett's multiple comparisons test, two-way ANOVA Šídák's multiple comparisons test, and uncorrected Fisher's LSD test, and the Wald test using the Benjamini-Hochberg method. Schematic figures were prepared with http://biorender.com.

## Online supplemental material

Fig. S1 (related to Fig. 1) shows N-cad expression, depletion, and controls for PHGG migration assays. Fig. S2 (related to Figs. 3 and 4) shows role of intercellular N-cad homotypic interactions and the importance of N-cad for catenin localization. Fig. S3 (related to Fig. 5) shows N-cad endocytosis, recycling, and surface levels in leader and follower cells. Fig. S4 (related to Fig. 6) shows YAP1 signaling and wound healing gene expression is increased in leader cells. Fig. S5 (related to Fig. 8) shows YAP1/TAZ regulates PHGG migration and N-cad endocytosis. Video 1 (related to Fig. 1) shows control and N-cad shRNA spheroid migration on laminin. Video 2 (related to Fig. 6) shows migrating leader and follower cells on neurons or laminin. Table S1 (related to Fig. 1) shows mRNA expression levels of cell adhesion receptors in patient-derived PHGGs. Table S2 (related to Fig. 1) shows RNA sequencing results comparing control and N-cad shRNA cells. Table S3 (related to Fig. 6) shows RNA sequencing results comparing leader and follower cells.

## Data availability

The data are available from the corresponding author upon reasonable request. The data underlying Table S1 and Fig. S1 E are openly available in R2: Genomics Analysis and Visualization, Mixed Pediatric PDX-Olson at http://r2.amc.nl. The data underlying Tables S2 and S3 are openly available in Dryad at https://doi.org/10.5061/dryad.4tmpg4fj7.

## Acknowledgments

We are very grateful to Emily J. Girard, Fiona Pakiam, and Shelli Morris at Seattle Children's Research Institute for their assistance in providing cells, reagents, and technical guidance during this study. We also thank Lena Schroeder, Jin Meng, Peng Guo, and Julien Dubrulle in the Cellular Imaging and Bioinformatics Shared Resources at Fred Hutchinson Cancer Center for imaging and analysis assistance, Flow Cytometry core staff for cell sorting instruction, Genomics core staff at Benaroya Research Institute for sequencing, and Luna Yu for computational assistance. We thank Saurav Kumar, Amanda Stainer, Liesje Steenkiste, Laura Arguedas-Jimenez, Chris Simpkins, Jay Sarthy, Mark Headley, David Helfman, Soo Young Kim, and other colleagues for discussions and comments on the manuscript.

This research was supported by the Fred Hutch Interdisciplinary Training in Cancer Research Program, institutional funds from Fred Hutch and the shared resources of the Fred Hutch/University of Washington/Seattle Children's Cancer Consortium (P30 CA015704).

Author contributions: Conceptualization, Formal Analysis, Investigation, Methodology, Validation, Visualization, Writing—

Original Draft Preparation, Writing—Review & Editing and Funding acquisition, D. Kim and J.A. Cooper; Resources, J.M. Olson and J.A. Cooper; Supervision, J.M. Olson and J.A. Cooper.

Disclosures: The authors declare no competing interests exist.

Submitted: 11 January 2024

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

# Supplemental material

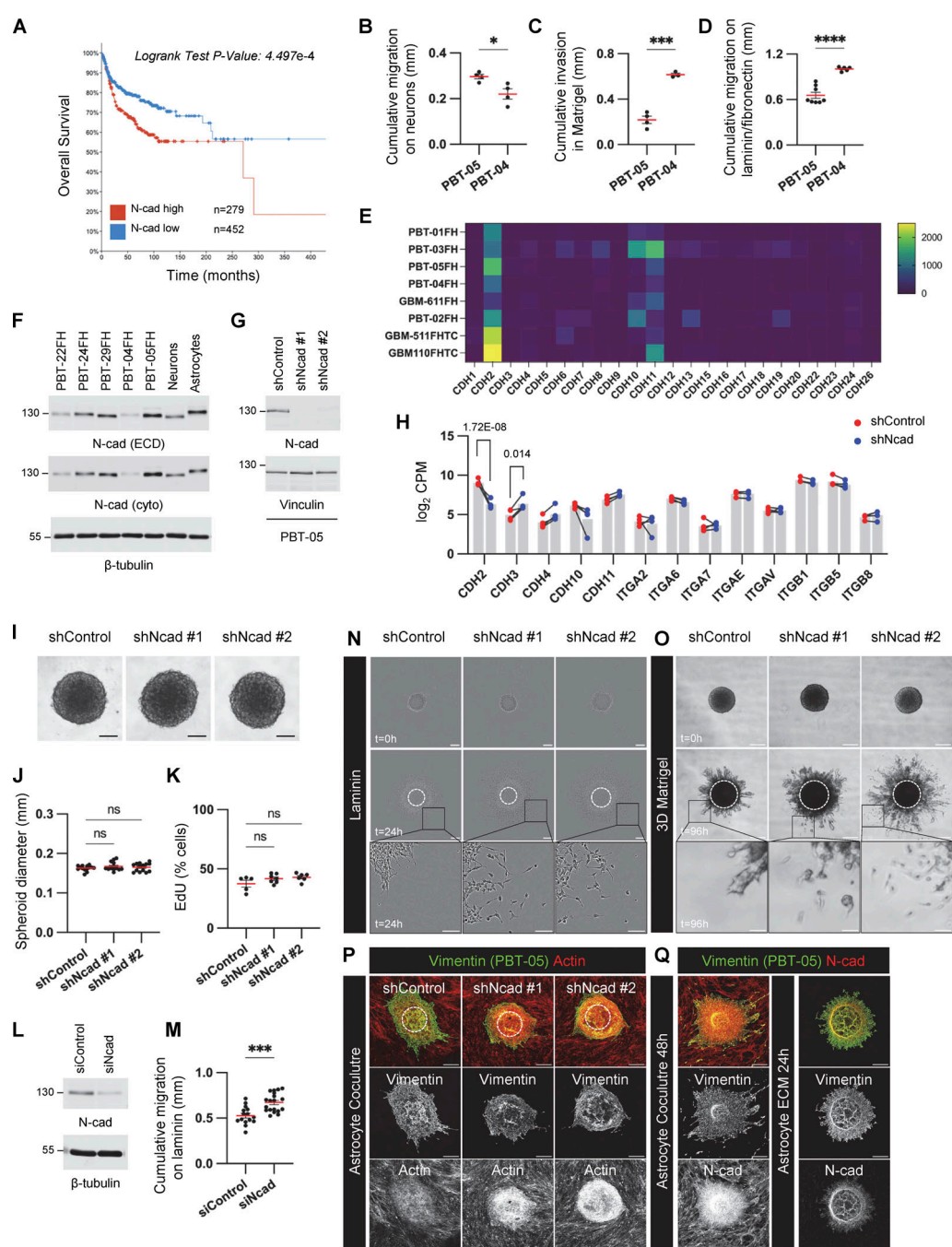

Figure S1.  **N-cad expression, depletion, and controls for PHGG migration assays. (A)** Overall survival of pediatric brain tumor patients with high or low N-cad mRNA expression. Data collected from Pediatric Brain Tumor Atlas (PBTA) (http://pedcbioportal.kidsfirstdrc.org, PedcBioPortal, Open Pediatric Brain Tumor Atlas, v22). **(B–D)** Migration or invasion distance of PBT-04 and PBT-05 cells in mouse cerebellar neurons, Matrigel, and laminin/fibronectin. Unpaired t tests. *P < 0.05, ***P < 0.001, ****P < 0.0001. **(E)** mRNA expression levels of cadherins in patient-derived PHGG tissues and cells (https://r2.amc.nl, R2: Genomics Analysis and Visualization, Mixed Pediatric PDX-Olson). **(F)** N-cad protein expression level in DMG (PBT-22FH, PBT-24FH, and PBT-29FH) and PHGG (PBT-04FH and PBT-05FH) cell lines and mouse cerebellar neurons and mouse astrocytes. β-Tubulin is shown as a loading control. **(G)** Western blots for control or N-cad shRNAs in PBT-05 cells. Vinculin is shown as a loading control. **(H)** RNA levels of N-cad and other cell adhesion proteins in control or N-cad shRNA PBT-05 cells. Mean CPM (counts per million) from four biological replicates. Lines connect paired samples. Adjusted P values from the Wald test using the Benjamini-Hochberg method. **(I–K)** N-cad does not regulate spheroid formation or cell proliferation. **(I)** Representative images of spheroids. Scale bar, 100 µm. **(J)** Spheroid diameter at start of migration experiment. Ordinary one-way ANOVA Holm-Šídák's multiple comparisons test. ns, not significant. N = 13–14 spheroids, four experiments. **(K)** EdU pulse labeling. Ordinary one-way ANOVA Holm-Šídák's multiple comparisons test. ns, not significant. N = 5–7 spheroids, three experiments. **(L and M)** Western blots and migration distance on laminin of control and N-cad siRNA-treated PBT-05 cells. Unpaired t test. ***P < 0.001. N = 15–19 spheroids, 4 experiments. **(N and O)** N-cad depletion increases single-cell migration. Representative images of migrating control or N-cad shRNA spheroids on laminin or 3D Matrigel. Scale bars, 200 or 50 µm (inset). **(P and Q)** Controls for migration on astrocytes and astrocyte ECM. **(P)** Cell migration on mouse astrocytes. PBT-05 cells were identified with human-specific anti-vimentin antibodies. PBT-05 cell and astrocyte actin were detected with phalloidin. **(Q)** Decellularized astrocyte cultures lack N-cad. Scale bars, 200 µm. Error bars indicate mean ± SEM. Source data are available for this figure: SourceData FS1.

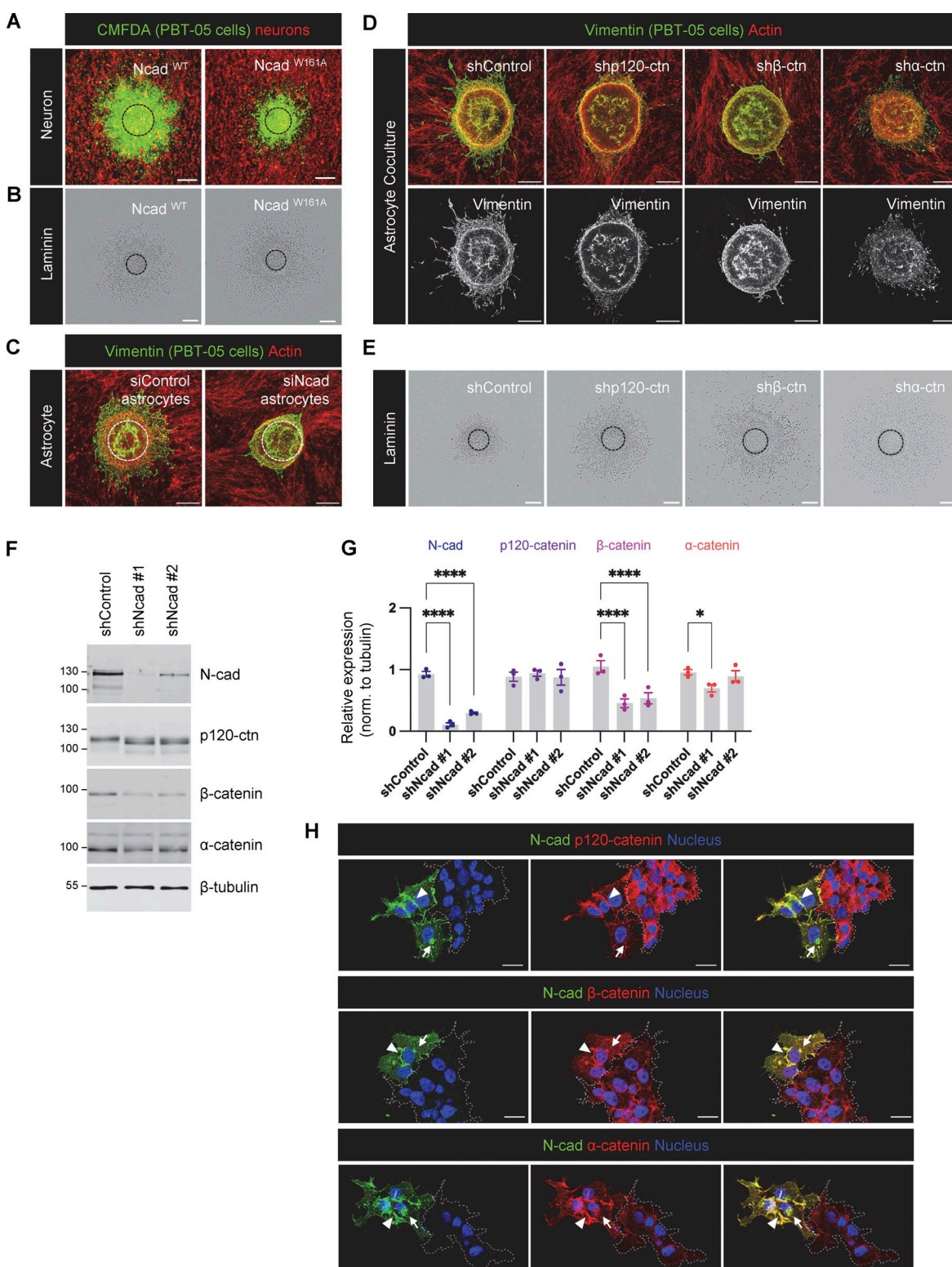

Figure S2. **Role of intercellular N-cad homotypic interactions and importance of N-cad for catenin localization. (A and B)** Representative images of NcadWT and NcadW161A expressing PBT-05 cells migrating on cerebellar neurons or laminin. **(C)** Representative images of PBT-05 cells migrating on control or N-cad siRNA-treated mouse astrocytes. Scale bars, 200 μm. **(D and E)** Images of PBT-05 cells expressing control, p120-catenin (p120-ctn), β-catenin (β-ctn), or α-catenin (α-ctn) shRNAs, migrating on mouse astrocytes or laminin. Scale bars, 200 μm. **(F and G)** N-cad, p120-catenin β-catenin, and α-catenin protein levels in control and N-cad-depleted cells. Protein expression levels were normalized to β-tubulin as a loading control. Two-way ANOVA uncorrected Fisher's LSD test. N = 3 Western blots. Error bars show mean ± SEM. *P < 0.05, ****P < 0.0001. **(H)** Localization of p120-catenin, β-catenin, and α-catenin in control or N-cad-depleted cells. Dashed lines indicate clusters of N-cad-depleted cells. Arrowheads indicate cell-cell junctions and arrows indicate intracellular vesicles. Scale bars, 20 μm. Source data are available for this figure: SourceData FS2.

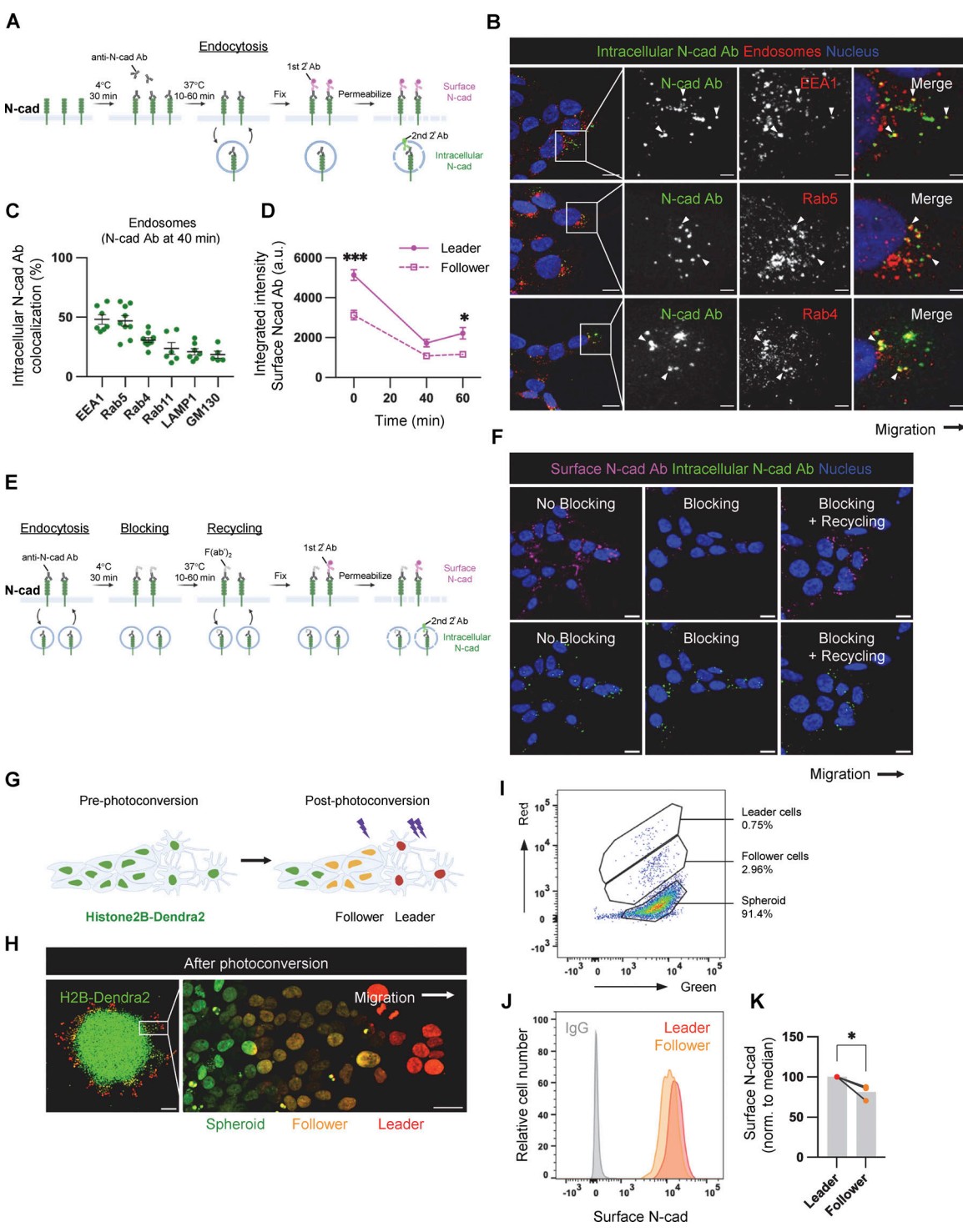

Figure S3.  **N-cad endocytosis, recycling, and surface levels in leader and follower cells. (A)** Schematic diagram of the N-cad antibody (Ab) internalization assay. Cells were labeled with N-cad ECD Ab in the cold, warmed for various times to allow internalization, fixed, and surface and internalized Ab detected with different fluorescent secondaries. **(B)** Immunofluorescence detection of internalized N-cad Ab with EEA1, Rab5, or Rab4 in leader cells after 40 min internalization. Arrowheads indicate colocalization. Scale bars, 10 or 2 μm (inset). **(C)** Object-based colocalization quantification of internalized N-cad Ab with EEA1, Rab5, Rab4, Rab11, LAMP1, or GM130. N = 5–10 spheroids, three experiments. **(D)** Surface N-cad Ab increases between 40 and 60 min; a.u., arbitrary units. Two-way ANOVA Šídák's multiple comparisons test multiple comparisons test. *P < 0.05, ***P < 0.001. N = 3–4 spheroids, three experiments. **(E and F)** Schematic and representative images of N-cad recycling assay. Cells were incubated with N-cad ECD antibody in the cold and warmed for 40 min to allow internalization. N-cad Ab remaining on the surface was blocked in the cold with excess F(ab')₂. Cells were then warmed to allow recycling before fixation and detection of surface and internalized Ab with different fluorescent secondaries. Scale bars, 10 μm. **(G and H)** Schematic diagram and representative images showing photoconversion of histone H2B-Dendra2 in leader and follower cells. **(I–K)** Flow cytometry. **(I)** Definition of leader, follower and spheroid cells according to extent of photoconversion. **(J)** N-cad surface intensity histograms. **(K)** Normalized median N-cad fluorescent intensity. Paired t test. *P = 0.0374. N = 3 experiments. Error bars show mean ± SEM.

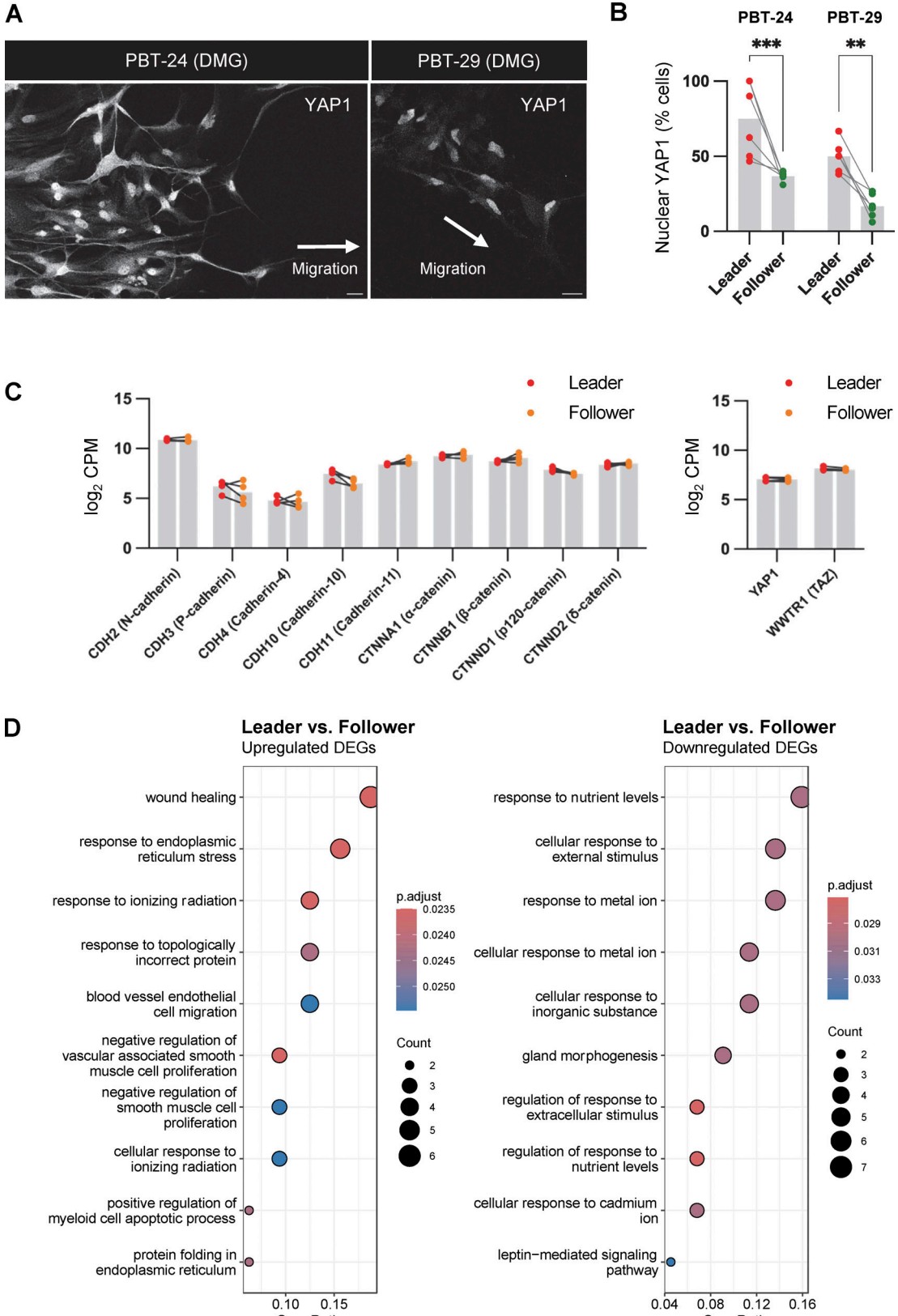

Figure S4. **YAP1 signaling and wound healing gene expression is increased in leader cells. (A and B)** Representative images and quantification of YAP1 localization in leader and follower cells in PBT-24 and PBT-29 DMG cells migrating on Matrigel for 72 h. Scale bars, 20 µm. *N* = 5 spheroids, three experiments. Two-way ANOVA Šídák's multiple comparisons test. Error bars indicate ±SEM. **P < 0.01, ***P < 0.001. **(C)** RNASeq analysis of cadherin, catenin, YAP1, and TAZ expression in leader and follower cells. **(D)** Gene ontology functional enrichment analysis of differentially expressed genes (DEGs) between leader and follower cells.

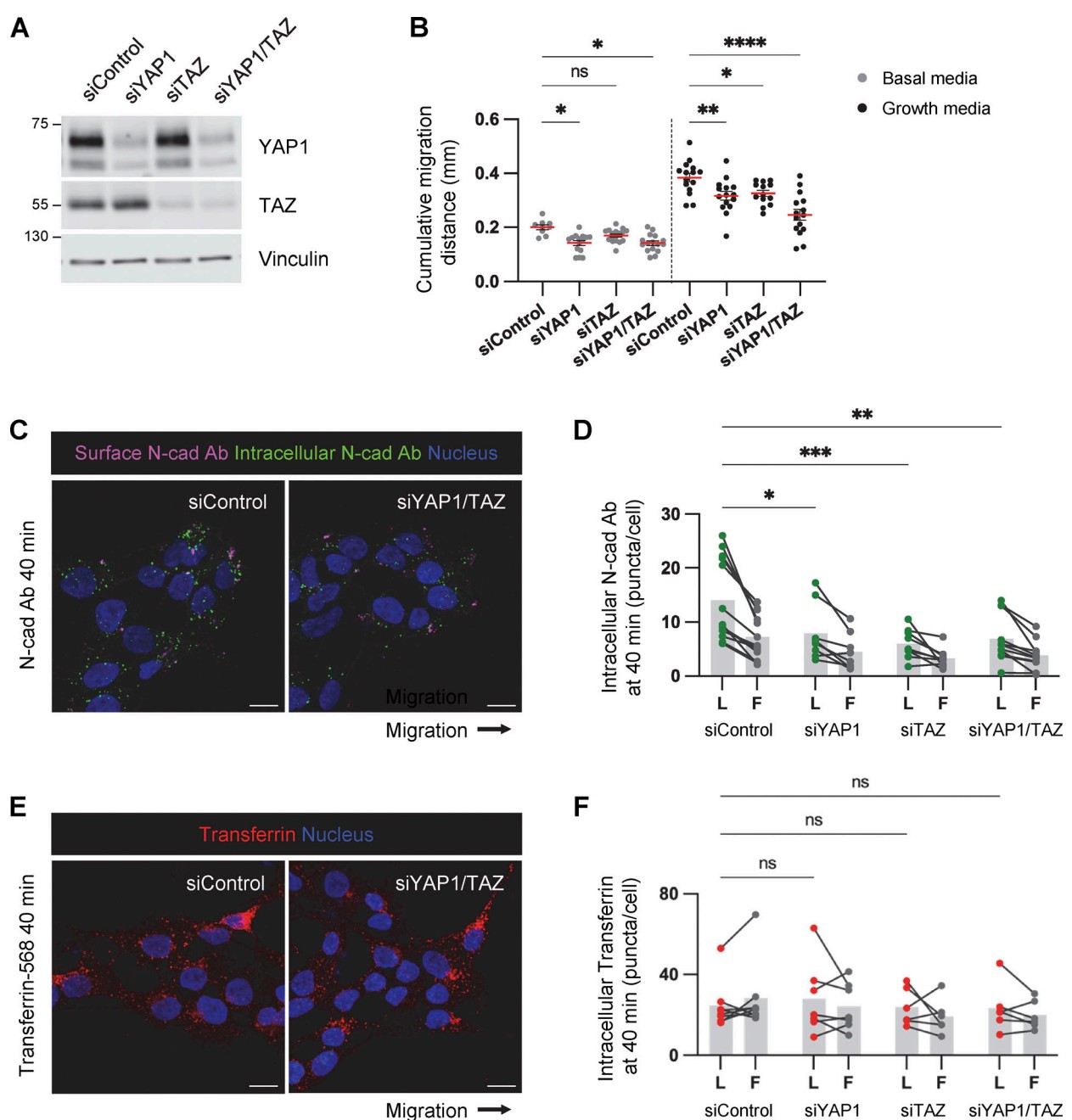

Figure S5. **YAP1/TAZ regulates PHGG migration and N-cad endocytosis. (A)** Western blot analysis of control, YAP1, TAZ, or YAP1/TAZ siRNAs in PBT-05 cells. Vinculin is shown as a loading control. **(B)** Cumulative migration distance on laminin for 24 h. Ordinary one-way ANOVA Šídák's multiple comparisons test. N = 9–16 spheroids, three experiments. Error bars indicate mean ± SEM. **(C and D)** Representative images and quantification of internalized N-cad Abs at 40 min incubation. N = 4–13 spheroids, three experiments. **(E and F)** Representative images and quantification of internalized Alexa Fluor 568-conjucated transferrin at 40 min incubation. N = 5–7 spheroids, two experiments. **(D and F)** Two-way ANOVA Šídák's multiple comparisons test. *P < 0.05, **P < 0.001, ***P < 0.001, ****P < 0.0001. ns, not significant. Source data are available for this figure: SourceData FS5.

Video 1. **N-cad inhibits PHGG cell migration and cell dissociation.** Control or N-cad shRNA PBT-05 spheroids were imaged every 5 min for 24 h during migration on laminin. Images were captured by Incucyte. Scale bars, 200 µm. Playback speed, 15 frames per second.

Video 2.  **Leader and follower cells interconvert during PHGG migration.** Histone H2B-Dendra2 expressing PBT-05 cells migrating on neurons or laminin. Time-lapse images were taken every 15 min for 16 h. Images were captured by spinning disk confocal microscope. Scale bars, 20 µm. Playback speed, 15 frames per second.

**Provided online are Table S1, Table S2, and Table S3. Table S1 shows the cell adhesion receptors in PHGGs. Table S2 shows RNA sequencing results comparing control and N-cad shRNA cells. Table S3 shows RNA sequencing results comparing leader and follower cells.**

