## [Peer Review File · The Journal of Cell Biology]

N-cadherin dynamically regulates pediatric glioma cell migration in complex environments

Dayoung Kim, James Olson, and Jonathan Cooper

Corresponding Author(s): Jonathan Cooper, Fred Hutchinson Cancer Center and Dayoung Kim, Fred Hutchinson Cancer Center

Review Timeline:

Submission Date:	2024-01-11
Editorial Decision:	2024-02-08
Revision Received:	2024-02-20

Monitoring Editor: Sandrine Etienne-Manneville

Scientific Editor: Andrea Marat

Transaction Report:

DOI: <https://doi.org/10.1083/jcb.202401057>

Revision 0

Review #1

1. Evidence, reproducibility and clarity:

Evidence, reproducibility and clarity (Required)

Kim et al review

In this manuscript, Kim and colleagues describe the role of N-Cadherin during pediatric glioma migration. They compare cell lines that have similar transcripts but different levels of N-Cadherin protein and find that N-Cadherin levels influence the route of migration - whether it be on ECM or other tissues. They also describe molecular feedback between N-Cadherin and YAP in leader vs follow cells of their systems.

The data are clear, well presented, and convincing; and the conclusions described by the manuscript are mostly justified. My major criticism of the manuscript is that the line of questioning undertaken does not appear well justified. At many points, I was left asking "but why are they doing this?" and I could not understand the rationale for some of the experiments that were performed (even if they were performed well). The manuscript opens by validly describing how gliomas are highly invasive, poorly understood and that N-Cadherin was highly expressed in comparison to other adhesion proteins. This opened the path for the questions and experiments performed that contributed to Figures 1-3, which I thought were interesting. From there on, I found the logic of the story unclear and poorly justified. For example, I do not know why leader and follower cells were justified - when it had nothing to do with N-Cadherin which was the focus of the work prior. And then, having rightly concluded in Figure 4 that the data suggested that leader and follower cells dynamically exchange positions rather than being pre-determined, they went onto further figures focusing on differences between leader and follower cells, which left me quite confused.

I am likewise confused by the model proposed in that, they authors describe that the difference between leader and follower cells contributes to a nuclear YAP/N-Cad endocytosis feedback loop that feeds into the speed of migration. Yet, the authors describe earlier that leader and follower cells frequently exchange positions, with no evidence that they are pre-determined. How do the authors square these seemingly conflicting points? And further, what is the relevance of this to understanding the differing modes of migration (on ECM or other tissues)?

On this issue, I suggest authors re-consider whether the order of figures or logic of the story is appropriate (perhaps consider moving some figures to supplement?), and to clearly justify in the text the elements that are being addressed.

Overall, I think the messaging, logic and justification could be use significant improvement; the experiments however are well performed, and the figures are very clear and nicely presented, and I don't have any qualms about them.

****Minor Comments****

- Not required, but the authors may wish to consider putting $t=0$ pictures of the experiments in the supplement as supportive evidence for the circles of the initial seeding location they show in Fig 1.
- I assume the title of the second results section should say "migration speed" rather than "speed migration"
- Fig. 4D - Are both example cell pictures leaders? If so, I'm not sure why two have been provided; I'm guessing the bottom set are supposed to be follower cells. If so, please label as appropriate. (And if not, a representative set of pictures from a follower cell should be provided).
- Figure 5 Legend - the title of this figure is too definitive, and exaggerates further than the main text does, which was correct in saying that the experiments only suggest that N-Cadherin endocytosis might regulate the localisation of β -catenin and p120-catenin. Probably I would go further and say that there is no experimental evidence provided that even suggests that in the first place, and that this is a hypothesis that remains to be tested. The authors should inhibit endocytosis specifically (rather than just depleting N-Cad) and see the effect, to justify their conclusion.

2. Significance:

Significance (Required)

The manuscript provides a characterised of invasive glioma migration that was previously lacking. It also provides interesting observations related to the role of N-Cadherin for different modes of migration (on ECM or on tissues) that will be of interest for further exploration. It makes a good advance in terms of addressing a highly invasive cell type that has poor prognosis. I anticipate that now this initial characterisation has been performed, authors and others will be interested in gaining a deeper understanding as to how these two modes of migration are controlled, how there might be interplay between them and how such findings contribute to its highly invasive nature.

I have expertise in collective cell migration and directed cell migration.

3. How much time do you estimate the authors will need to complete the suggested revisions:

Estimated time to Complete Revisions (Required)

(Decision Recommendation)

Less than 1 month

4. Review Commons values the work of reviewers and encourages them to get credit for their work. Select 'Yes'

below to register your reviewing activity at Web of Science Reviewer Recognition Service (formerly Publons); note that the content of your review will not be visible on Web of Science.

Yes

Review #2

1. Evidence, reproducibility and clarity:

Evidence, reproducibility and clarity (Required)

****Summary****

In the submitted manuscript, Kim et al. describe various aspects of N-cadherin function in the collective migration of PBT-05 cells, a pediatric high-grade glioma line, on laminin, 3D-matrigel, neurons or astrocytes. N-cadherin promotes the collective migration on neurons or astrocytes, whereas it suppresses the migration on laminin or 3D-matrigel. The authors also show that N-cadherin is actively internalized and recycled in the leader, but not follower, cells of the collective, which induce the nuclear accumulation of YAP/TAZ proteins. YAP/TAZ proteins are shown to regulate the collective migration.

****Major comments****

1. In Fig. 1G, N-cadherin knockdown seems to affect the distribution of astrocytes. The authors should stain a marker for astrocytes, instead of actin, and the red alone images should be provided.
2. The colocalization between N-cadherin and Rab11 may not be high in Figs. 4F and S2B. It is unclear whether the majority of the internalized N-cadherin is recycled to the plasma membrane. In Fig. S2B, the internalized N-cadherin may be located mainly at the early endosomes before transported to the recycling endosomes (Is it 20 min after the N-cadherin antibody internalization?). First, the authors should analyze the colocalization between the N-cadherin and Rab11 at 30-40 min after the internalization. If the colocalization with Rab11 would be still low at that time point, some of the internalized N-cadherin might be degraded in the lysosomes. To test this possibility, the authors should analyze the colocalization between N-cadherin and LAMP1 under the treatment with a lysosome inhibitor.
3. When N-cadherin is depleted, dissociated single cells are increased, but these cells are not well characterized. A high magnification image of the dissociated single cells is required. In addition,

the migration speed of the dissociated single cells should be measured.

4. In Fig. S2D, treatment with Pitstop-2 alone or Dyngo-4a alone is required. Dynamin is also involved in clathrin-independent endocytosis and N-cadherin is reported to be internalized via caveolin-1-mediated endocytosis as well as clathrin-mediated during neuronal migration. It would be better to clarify which type of endocytosis occurs in the leader cells.

5. In Fig. 2, N-cadherin depletion disturbs the migration directionality. Is this a result from disruption of cell polarity? To test this, the position of centrosome or Golgi or lamellipodia in the leader cells should be analyzed. (OPTIONAL)

6. I cannot understand the significance of Fig. 5F and 5G. If the authors would speculate that alpha- and beta-Catenins may transduce the intracellular signaling from the internalized N-cadherin, the authors should perform the knockdown experiments of the Catenins and analyze whether it may affect the nuclear accumulation of YAP/TAZ. (OPTIONAL)

****Minor comments****

7. The quantitative data is required in Fig. 5E.

8. Vinculin is associated with the cadherin-catenin complex and it may not be a good loading control (Fig. 3C and 3L).

****Referees cross-commenting****

I totally agree with the other Reviewers' comments and evaluation. As the reviewer-1 pointed out, I also think the experiments are well performed, but it would lack logic at least in part (see my comment-6). In addition, as the reviewer-3 pointed out, the linking mechanism of N-cadherin homophilic interaction with YAP/TAZ signaling is important to improve this manuscript

2. Significance:

Significance (Required)

Strength

N-cadherin has multiple function in cancer and neuronal migration, and both positive and negative effects of N-cadherin on cancer cell migration have been reported. In this regard, different behaviors of N-cadherin in the leader and follower cells of the collective are interesting and may explain the controversial previous results.

Limitation

This study reveals various aspects of N-cadherin function in the collective migration of the glioma cell line, but it is unclear whether these findings are applied to pediatric high-grade gliomas in vivo.

Thus, this study is a potentially important and informative to cell biologists and researchers in cancer biology, although this reviewer also found several weak points that should be improved.

3. How much time do you estimate the authors will need to complete the suggested revisions:

Estimated time to Complete Revisions (Required)

(Decision Recommendation)

Between 3 and 6 months

Yes

Review #3

1. Evidence, reproducibility and clarity:

Evidence, reproducibility and clarity (Required)

In this manuscript, the authors explore the role of N-cadherin in the migratory/infiltrative behavior of human pediatric brain tumor cells, in light of their surrounding microenvironment. Their in-depth phenotype analysis allows to document the behavior of migrating cells and revisit the concept of leading/follower migratory cells (somehow more commonly applied to endothelial cells).

They suspected that the YAP/TAZ pathway might modulate N-cadherin endocytosis and vice versa, using imagery-based cell tracking.

****Major comments:****

1. To control for co-culture models, migration should be evaluated on decellularized matrices from astrocyte and neuron cultures.
2. N-cadherin was stably knocked down with shRNA, which raises the question of

adaptative/compensatory mechanisms. First, one could ask what happen in knockout conditions. Similarly, transient siRNA-mediated silencing might help to strengthen the findings. Second, is there any impact of Ncad knock down on alternate adhesive receptors (either cell-cell or cell-ECM). This should be verified with bulk RNAseq.

3. It would be interesting to evaluate the impact of N-cadherin/N-cadherin homotypic interactions on YAP/TAZ signaling, using for instance N-cad-coated surface.

4. along this line, the impact of mechanical cues (stiffness, 2D vs 3D) is not explored.

****Minor comments:****

1. Levels of N-cadherin expression in normal Astro and Neurons to compare with pediatric brain cancer cells (S1C)

2. Low versus high density culture conditions should be controlled and its further impact on the YAP/Ncad endocytosis route should be supported experimentally, or to be omitted from their proposed model.

2. Significance:

Significance (Required)

This paper presents robust image analysis of human pediatric brain tumor migration in the context of the different microenvironment that they might encounter (matrices, neurons, astrocytes).

This study brings new concepts on the way N-cadherin might contribute to tumor cell migratory behavior based on the nature of the interactions in which N-cadherin is involved.

As a limitation, it remains unclear the mechanism by which N-cadherin endocytosis is driven.

3. How much time do you estimate the authors will need to complete the suggested revisions:

Estimated time to Complete Revisions (Required)

(Decision Recommendation)

Between 3 and 6 months

4. Review Commons values the work of reviewers and encourages them to get credit for their work. Select 'Yes'

below to register your reviewing activity at Web of Science Reviewer Recognition Service (formerly Publons); note that the content of your review will not be visible on Web of Science.

Yes

Manuscript number: RC-2023-01952R

Corresponding author(s): Jonathan A. Cooper

1. General Statements

We appreciate the insightful reviewer comments. Both reviewers alluded to the logical lack of connection between two themes in the original paper. Specifically, we showed that N-cad differentially regulates migration in different environments, and that leader and follower cells differ phenotypically, but did not connect the two themes. In this revised version, we've performed additional experiments and undertaken a comprehensive reorganization of both the manuscript and figures. The major changes are outlined below:

- Figure 4 A-C has been moved to Figure 6 F-H.
- Figure 5 has been moved to Figure S3 F-H.
- Figure 6 F has been moved to Figure 7 A.
- Figure 6 G-H have been moved to Figure 7 D-E.
- Figure 6 I-J have been moved to Figure S5 A-B.
- Figure 7 C-F have been moved to Figure S5 C-F.
- Added transcriptome profiling of control and N-cad-depleted cells and of leader and follower cells (Figures 6 E, S1 H and S4 C-D, Tables S2 and S3).

We have incorporated additional figures (Figure 4 and 5 in the revised manuscript) to support the idea that the amount of N-cad at the cell surface is regulated by endocytic recycling, thereby stimulating glioma migration in the different local environments. Furthermore, our new findings showed that YAP1/TAZ regulates the surface level of N-cad during glioma migration (Figure 8). We trust that these additions contribute to the clarity and robust justification of our paper.

Similar to other types of tumors, our findings revealed that pediatric high-grade gliomas migrate collectively, possibly contributing to a more aggressive invasion than single cells. In this study, we found that N-cad mediates this collective glioma migration. Interestingly, within these migrating groups, leader and follower cells dynamically interchange positions during migration, accompanied by changes their phenotypic characteristics. This suggests that differences in phenotypes, including N-cad recycling, proliferation and YAP activation, may be predominantly regulated by cell-extrinsic factors rather than being predetermined by genetic or epigenetic factors. Moreover, our new RNA-sequencing results indicate minimal difference between leader and follower cells, except for the upregulation of YAP response and wound healing migration genes in leader cells. Although genomic alterations still possibly encode the leader-follower exchange, our findings strongly suggest that the activation of YAP1 and glioma migration are regulated by the cellular context, specifically where cells are located within the group.

Contrary to our initial findings suggesting a positive feedback loop between N-cad endocytosis and nuclear YAP1, our revised data indicates that nuclear YAP appears to be independent of N-cad. We observed that homotypic interactions with N-cad present in the surrounding environment, such as neurons (Figure 6 C-D) or N-cad extracellular domain-coated surface (Figure 7 B-C), did not affect nuclear YAP1. However, YAP1/TAZ depletion decreased N-cad expression and altered its localization at the surface (Figure 8). This leads us to propose a revised model where nuclear YAP1 stimulates surface N-cad, thereby facilitating the distinct modes of migration on ECM and neurons (Figure 8 I).

Reviewer #1 (Evidence, reproducibility and clarity (Required)):

In this manuscript, Kim and colleagues describe the role of N-Cadherin during pediatric glioma migration. They compare cell lines that have similar transcripts but different levels of N-Cadherin protein and find that N-Cadherin levels influence the route of migration - whether it be on ECM or other tissues. They also describe molecular feedback between N-Cadherin and YAP in leader vs follow cells of their systems. The data are clear, well presented, and convincing; and the conclusions described by the manuscript are mostly justified. My major criticism of the manuscript is that the line of questioning undertaken does not appear well justified. At many points, I was left asking "but why are they doing this?" and I could not understand the rationale for some of the experiments that were performed (even if they were performed well). The manuscript opens by validly describing how gliomas are highly invasive, poorly understood and that N-Cadherin was highly expressed in comparison to other adhesion proteins. This opened the path for the questions and experiments performed that contributed to Figures 1-3, which I thought were interesting. From there on, I found the logic of the story unclear and poorly justified. For example, I do not know why leader and follower cells were justified - when it had nothing to do with N-Cadherin which was the focus of the work prior. And then, having rightly concluded in Figure 4 that the data suggested that leader and follower cells dynamically exchange positions rather than being pre-determined, they went onto further figures focusing on differences between leader and follower cells, which left me quite confused. I am likewise confused by the model proposed in that, they authors describe that the difference between leader and follower cells contributes to a nuclear YAP/N-Cad endocytosis feedback loop that feeds into the speed of migration. Yet, the authors describe earlier that leader and follower cells frequently exchange positions, with no evidence that they are pre-determined. How do the authors square these seemingly conflicting points? And further, what is the relevance of this to understanding the differing modes of migration (on ECM or other tissues)? On this issue, I suggest authors re-consider whether the order of figures or logic of the story is appropriate (perhaps consider moving some figures to supplement?), and to clearly justify in the text the elements that are being addressed. Overall, I think the messaging, logic and justification could

Full Revision

be use significant improvement; the experiments however are well performed, and the figures are very clear and nicely presented, and I don't have any qualms about them.

We appreciate your insightful comments, recognizing the need for logical and justifiable improvements in certain sections of our previous manuscript. Please see Section 1, General Statements, for an explanation of changes made.

Minor Comments

1. Not required, but the authors may wish to consider putting $t=0$ pictures of the experiments in the supplement as supportive evidence for the circles of the initial seeding location they show in Fig 1.

We provide the $t=0$ images in Figure S1 N and O.

2. I assume the title of the second results section should say "migration speed" rather than "speed migration"

The new title of the second results section is "N-cad stimulates and inhibits migration through intercellular homotypic interaction".

3. Fig. 4D - Are both example cell pictures leaders? If so, I'm not sure why two have been provided; I'm guessing the bottom set are supposed to be follower cells. If so, please label as appropriate. (And if not, a representative set of pictures from a follower cell should be provided).

We have enhanced the clarity of the labels. We provide representative high magnification images of leader and follower cells. The updated figure can be found in Figure 5 A.

4. Figure 5 Legend - the title of this figure is too definitive, and exaggerates further than the main text does, which was correct in saying that the experiments only suggest that N-Cadherin endocytosis might regulate the localisation of b-catenin and p120-catenin. Probably I would go further and say that there is no experimental evidence provided that even suggests that in the first place, and that this is a hypothesis that remains to be tested. The authors should inhibit endocytosis specifically (rather than just depleting N-Cad) and see the effect, to justify their conclusion.

We appreciate your points and concerns. Following your earlier suggestion, we have moved the figure to the supplementary section (Figure S3 F-H). Moreover, we have addressed the reciprocal regulation of N-cad and catenins by knocking down p120-, β - or α -catenin. Our new findings showed that p120-, β - or α -catenin depletion decrease the amount of N-cad at the cell surface, not steady-state protein level, resulting in decreased migration on astrocytes but increased migration on ECM (see Figure 4). These findings support the idea that catenins play a role in glioma migration according to the environment by altering surface N-cad level. With that, we updated the figure title to "Catenins regulate N-cad surface levels to stimulate or inhibit migration."

Full Revision

Reviewer #1 (Significance (Required)):

The manuscript provides a characterised of invasive glioma migration that was previously lacking. It also provides interesting observations related to the role of N-Cadherin for different modes of migration (on ECM or on tissues) that will be of interest for further exploration. It makes a good advance in terms of addressing a highly invasive cell type that has poor prognosis. I anticipate that now this initial characterisation has been performed, authors and others will be interested in gaining a deeper understanding as to how these two modes of migration are controlled, how there might be interplay between them and how such findings contribute to its highly invasive nature.

I have expertise in collective cell migration and directed cell migration.

Reviewer #2 (Evidence, reproducibility and clarity (Required)):

Summary

In the submitted manuscript, Kim et al. describe various aspects of N-cadherin function in the collective migration of PBT-05 cells, a pediatric high-grade glioma line, on laminin, 3D-matrigel, neurons or astrocytes. N-cadherin promotes the collective migration on neurons or astrocytes, whereas it suppresses the migration on laminin or 3D-matrigel. The authors also show that N-cadherin is actively internalized and recycled in the leader, but not follower, cells of the collective, which induce the nuclear accumulation of YAP/TAZ proteins. YAP/TAZ proteins are shown to regulate the collective migration.

Thank you for the comments. Please see Section 1, General Statements, for a summary of changes made. Please also note that our new experiments failed to show that N-cad levels or traffic regulate YAP/TAZ nuclear accumulation. Rather, YAP/TAZ are regulated by cell density independent of N-cad, and YAP/TAZ regulate N-cad protein levels and traffic independent of changes in N-cad RNA levels.

Major comments

1. In Fig. 1G, N-cadherin knockdown seems to affect the distribution of astrocytes. The authors should stain a marker for astrocytes, instead of actin, and the red alone images should be provided.

Astrocytes were cultured for 4 days to generate 3D scaffolds before adding the glioma spheroid, essentially as described (Gritsenko et al., *Histochem Cell Biol*, 2017). Co-cultures were stained for human-specific vimentin (glioma) or actin (glioma and astrocytes) (see Figure 1 G and separate channels in new Figure S1 P). There do not appear to be major changes in astrocyte organization outside the migration front, but we lack a way to stain for astrocytes specifically and cannot visualize astrocytes under the glioma cells. It remains possible that astrocytes may be affected differently by contact with control and N-cad-deficient glioma cells. However, we added a new experiment, assaying migration on decellularized astrocyte ECM. While N-cad stimulated migration on astrocytes it inhibited migration on astrocyte ECM (Figures 1 I and J and S1 Q). Thus N-cad stimulates glioma migration on astrocyte cells and not their ECM.

2. The colocalization between N-cadherin and Rab11 may not be high in Figs. 4F and S2B. It is unclear whether the majority of the internalized N-cadherin is recycled to the plasma membrane. In Fig. S2B, the internalized N-cadherin may be located mainly at the early endosomes before transported to the recycling endosomes (Is it 20 min after the N-cadherin antibody internalization?). First, the authors should analyze the colocalization between the N-cadherin and Rab11 at 30-40 min after the internalization. If the colocalization with Rab11 would be still low at that time point, some of the internalized N-cadherin might be degraded in the lysosomes. To test this possibility, the authors should analyze the colocalization between N-cadherin and LAMP1 under the treatment with a lysosome inhibitor.

Full Revision

At steady state, N-cad co-localized better with Rab5 than with Rab11 or LAMP1 (Figure 5 C-D). In kinetics experiments, N-cad antibodies were internalized for 40 min. They colocalized better with Rab5 or EEA1 than with Rab11 or LAMP1. When we allowed recycling for an additional 20 min, the surface level of N-cad antibodies partially recovered in leader cells more than follower cells (see Figures 5 G and S3 D). We tested whether treatment with lysosomal inhibitors would increase co-localization of N-cad with Rab11 in recycling endosomes. Surprisingly, however, Chloroquine or Bafilomycin A1 decreased the amount of internalized N-cad antibody in leader and follower cells, and long-term treatment did not increase total N-cad levels. Therefore, the fate of internalized N-cad in follower cells remains unclear.

3. When N-cadherin is depleted, dissociated single cells are increased, but these cells are not well characterized. A high magnification image of the dissociated single cells is required. In addition, the migration speed of the dissociated single cells should be measured.

We didn't quantify single cell migration because only a minority of cells separate from the collective even when N-cad is depleted. Therefore, we quantified migration directionality and speed for cells at or near the front of collective migration (Figure 2 D-I). We have updated the image of single cells, providing representative high-magnification images in Figure S1 N and O.

4. In Fig. S2D, treatment with Pitstop-2 alone or Dyngo-4a alone is required. Dynamin is also involved in clathrin-independent endocytosis and N-cadherin is reported to be internalized via caveolin-1-mediated endocytosis as well as clathrin-mediated during neuronal migration. It would be better to clarify which type of endocytosis occurs in the leader cells.

We have removed results showing inhibition of cell migration and N-cad endocytosis by Pitstop-2 and Dyngo-4a from the paper. Treatment with either chemical alone had much less effect on internalization or migration than adding both together (see figure below). This is hard to explain. Pitstop-2 should inhibit clathrin-coated pit formation and Dyngo-4a should inhibit clathrin and caveolin-mediated endocytosis. Caveolin-1 and 2 transcripts were not detected in our cells (Table S2). There may be some other form of clathrin-independent endocytosis. Interpretation is also challenging since these inhibitors will inhibit endocytosis of many receptors, not just N-cad. Accordingly, we have removed these results in the revised manuscript.

5. In Fig. 2, N-cadherin depletion disturbs the migration directionality. Is this a result from disruption of cell polarity? To test this, the position of centrosome or Golgi or lamellipodia in the leader cells should be analyzed. (OPTIONAL)

We elected not to perform this analysis.

6. I cannot understand the significance of Fig. 5F and 5G. If the authors would speculate that alpha- and beta-Catenins may transduce the intracellular signaling from the internalized N-cadherin, the authors should perform the knockdown experiments of the Catenins and analyze whether it may affect the nuclear accumulation of YAP/TAZ. (OPTIONAL)

We agree. In the initial manuscript, we showed that N-cad depletion altered the localization of p120-, β -, and α -catenin (previously shown in Figure 5 F-G). For better clarity and logic, these figures have been moved to Figure S2 H in the revised manuscript. Additionally, to test whether catenins regulate N-cad and YAP1, we depleted p120-, β -, or α -catenin using shRNA. We found that downregulation of p120-, β -, or α -catenin decreased N-cad surface levels, consequently slowing migration on astrocytes and stimulating migration on laminin (Figure 4). In other words, depleting catenins altered migration in parallel with the changes in N-cad surface level. Catenin depletion also increased single-cell dissociation, reduced the crowding of leader and follower cells, and increased nuclear YAP1 (see figure below). These findings suggest that the main role of p120-, β -, or α -catenin is to regulate surface N-cad. Since this result does not support a role for catenins transducing an N-cad signal to YAP1, we have not included it in the paper.

Minor comments

1. The quantitative data is required in Fig. 5E.

Quantitative data from three independent experiment are now presented in Figure S2 G.

2. Vinculin is associated with the cadherin-catenin complex and it may not be a good loading control (Fig. 3C and 3L).

The Western blot data has been updated and is now presented in new Figure 3 B and 3 F, with β -tubulin as a loading control.

****Referees cross-commenting****

I totally agree with the other Reviewers' comments and evaluation. As the reviewer-1 pointed out, I also think the experiments are well performed, but it would lack logic at least in part (see my comment-6). In addition, as the reviewer-3 pointed out, the linking mechanism of N-cadherin homophilic interaction with YAP/TAZ signaling is important to improve this manuscript.

We hope the revisions have improved the logical flow. We have also added new results showing that YAP/TAZ regulate N-cad protein levels and localization but not N-cad RNA. N-cad is not needed for cell density-dependent regulation of YAP1 localization. The model is shown in Figure 8 I.

Reviewer #2 (Significance (Required)):

Strength

N-cadherin has multiple function in cancer and neuronal migration, and both positive and negative effects of N-cadherin on cancer cell migration have been reported. In this regard, different behaviors of N-cadherin in the leader and follower cells of the collective are interesting and may explain the controversial previous results.

Limitation

This study reveals various aspects of N-cadherin function in the collective migration of the glioma cell line, but it is unclear whether these findings are applied to pediatric high-grade gliomas in vivo.

Thus, this study is a potentially important and informative to cell biologists and researchers in cancer biology, although this reviewer also found several weak points that should be improved.

Reviewer #3 (Evidence, reproducibility and clarity (Required)):

In this manuscript, the authors explore the role of N-cadherin in the migratory/infiltrative behavior of human pediatric brain tumor cells, in light of their surrounding microenvironment. Their in-depth phenotype analysis allows to document the behavior of migrating cells and revisit the concept of leading/follower migratory cells (somehow more commonly applied to endothelial cells). They suspected that the YAP/TAZ pathway might modulate N-cadherin endocytosis and vice versa, using imagery-based cell tracking.

Major comments

1. To control for co-culture models, migration should be evaluated on decellularized matrices from astrocyte and neuron cultures.

We thank for your suggestion. We tested glioma migration on astrocyte-derived decellularized matrices. The mouse astrocytes we used are known to produce various extracellular matrices with a composition similar to Matrigel, except for laminin $\alpha 5$. (Gritsenko et al., J Cell Sci, 2018). N-cad shRNA cells migrated faster on decellularized ECM than control (Figure 1 I-J and S1 Q). This result agrees with N-cad depletion increasing migration on ECM but is opposite to migration on astrocytes.

2. N-cadherin was stably knocked down with shRNA, which raises the question of adaptative/compensatory mechanisms. First, one could ask what happen in knockout conditions. Similarly, transient siRNA-mediated silencing might help to strengthen the findings. Second, is there any impact of Ncad knock down on alternate adhesive receptors (either cell-cell or cell-ECM). This should be verified with bulk RNAseq.

Transient knockdown with N-cad siRNA also increased migration on laminin-coated surface (Figure S1 L-M). Unfortunately, N-cad depletion with siRNA was short-lived, precluding its use for long-term assays, like coculture with neurons or astrocytes. To test whether there is any impact of N-cad knockdown on alternative adhesion receptors, we performed RNA-Seq (Figure S1 H, Table S2). We found N-cad depletion did not alter expression of other cell-cell and cell-ECM adhesive receptors except CDH3 (2.8-fold increase compared with 7-fold decrease in CDH2). Integrin expression was unchanged.

3. It would be interesting to evaluate the impact of N-cadherin/N-cadherin homotypic interactions on YAP/TAZ signaling, using for instance N-cad-coated surface.

We observed that the homotypic interaction of N-cad with surrounding neurons and astrocytes did not hinder the accumulation of nuclear YAP1 in leader cells (Figure 6 C-D). To further support the idea that N-cad does not directly regulate YAP1 signaling, we have now measured YAP1 localization in cells migrating over N-cad ECD. The new data confirms that N-cad does not directly regulate YAP1 localization (Figure 7 B-C).

4. along this line, the impact of mechanical cues (stiffness, 2D vs 3D) is not explored.

We appreciate your suggestion. It is possible that different mechanical and cytoskeletal cues between leader and follower cells affect YAP1 signaling. In this study, we would like to focus more on the role of N-cad-mediated cell adhesions in YAP signaling.

Minor comments

1. Levels of N-cadherin expression in normal Astro and Neurons to compare with pediatric brain cancer cells (S1C)

A new western blot analysis to show N-cad levels in DMG, PHGG and mouse cerebellar neurons and astrocytes has been added to Figure S1 F.

2. Low versus high density culture conditions should be controlled and its further impact on the YAP/Ncad endocytosis route should be supported experimentally, or to be omitted from their proposed model.

We previously used different size of micropattern discs to control low or high cell density. Smaller cell clusters, with more edge cells and hence fewer cell-cell interactions, had higher nuclear YAP1 (Figure 7 D-E). We have repeated this experiment, including N-cad ECD antibodies to measure N-cad endocytosis. Smaller cell clusters had higher N-cad antibody internalization (Figure 7 F). Together with our evidence that leader cells have higher YAP1 and more N-cad internalization than followers, and that YAP/TAZ knockdown inhibits N-cad internalization, these results high YAP/TAZ in leader cells regulates N-cad internalization.

Reviewer #3 (Significance (Required)):

This paper presents robust image analysis of human pediatric brain tumor migration in the context of the different microenvironment that they might encounter (matrices, neurons, astrocytes). This study brings new concepts on the way N-cadherin might contribute to tumor cell migratory behavior based on the nature of the interactions in which N-cadherin is involved. As a limitation, it remains unclear the mechanism by which N-cadherin endocytosis is driven.

We now discuss the limitations of the study as follows:

“The mechanisms by which YAP1 regulates N-cad levels and trafficking remain to be explored. YAP1 is widely expressed in human brain tumors and strongly associated poor survival. Leader cells expressed higher levels of YAP1-response and wound-healing gene transcripts, but transcript levels of N-cad and proteins known to regulate cadherin traffic, such as p120-catenin, Rab5/11 and Rac1, were similar. Therefore, N-cad is likely regulated at the level of protein synthesis or turnover. More endosomal N-cad recycled to the surface of leader than follower cells, implying that follower cells might divert more N-cad for lysosomal degradation, but our

Full Revision

attempts to interfere with N-cad endocytosis or degradation specifically were unsuccessful. Further understanding of the mechanism and function of N-cad recycling for glioma cell migration will require cargo-specific ways to selectively regulate endocytosis and recycling”.

February 8, 2024

RE: JCB Manuscript #202401057T

Dr. Jonathan A Cooper
Fred Hutchinson Cancer Center
Basic Sciences
1100 Fairview Ave N
Seattle, Washington 98109

Dear Dr. Cooper:

We thank the authors for their revised and improved manuscript. The two reviewers who have looked at the revised version consider that the manuscript is now suitable for publication in JCB. We are therefore pleased to inform you that the article "N-cadherin dynamically regulates pediatric glioma cell migration in complex environments " is in principle accepted for publication in JCB. However, the authors should answer the final concern raised by the reviewer 1, either by adding an immunofluorescence analysis of integrins and/or by discussing this point in the manuscript.

A. MANUSCRIPT ORGANIZATION AND FORMATTING:

- 1) Text limits: Character count for Articles is < 40,000, not including spaces. Count includes abstract, introduction, results, discussion, and acknowledgments. Count does not include title page, figure legends, materials and methods, references, tables, or supplemental legends.
- 2) Figures limits: Articles may have up to 10 main text figures.
- 3) Figure formatting: Scale bars must be present on all microscopy images, including inset magnifications. Molecular weight or nucleic acid size markers must be included on all gel electrophoresis. In order to accommodate readers with red-green color blindness, we request that you please change any red/green color schemes.
- 4) Statistical analysis: Error bars on graphic representations of numerical data must be clearly described in the figure legend. The number of independent data points (n) represented in a graph must be indicated in the legend. Statistical methods should be explained in full in the materials and methods. For figures presenting pooled data the statistical measure should be defined in the figure legends. Please also be sure to indicate the statistical tests used in each of your experiments (either in the figure legend itself or in a separate methods section) as well as the parameters of the test (for example, if you ran a t-test, please indicate if it was one- or two-sided, etc.). Also, if you used parametric tests, please indicate if the data distribution was tested for normality (and if so, how). If not, you must state something to the effect that "Data distribution was assumed to be normal but this was not formally tested."
- 5) Abstract and title: The abstract should be no longer than 160 words and should communicate the significance of the paper for a general audience. The title should be less than 100 characters including spaces. Make the title concise but accessible to a general readership.
- 6) Materials and methods: Should be comprehensive and not simply reference a previous publication for details on how an experiment was performed. Please provide full descriptions in the text for readers who may not have access to referenced manuscripts.
- 7) *All antibodies, cell lines, animals, and tools used in the manuscript should be described in full, including accession numbers for materials available in a public repository such as the Resource Identification Portal. Please be sure to provide the sequences for all of your primers/oligos and RNAi constructs in the materials and methods. You must also indicate in the methods the source, species, and catalog numbers (where appropriate) for all of your antibodies. Please also indicate the acquisition and quantification methods for immunoblotting/western blots.*
- 8) Microscope image acquisition: The following information must be provided about the acquisition and processing of images:
 - a. Make and model of microscope
 - b. Type, magnification, and numerical aperture of the objective lenses

- c. Temperature
- d. Imaging medium
- e. Fluorochromes
- f. Camera make and model
- g. Acquisition software
- h. Any software used for image processing subsequent to data acquisition. Please include details and types of operations involved (e.g., type of deconvolution, 3D reconstitutions, surface or volume rendering, gamma adjustments, etc.).

10) Supplemental materials: There are strict limits on the allowable amount of supplemental data. Articles may have up to 5 supplemental figures. Please also note that tables, like figures, should be provided as individual, editable files. A summary of all supplemental material should appear at the end of the Materials and methods section.

13) ORCID IDs: ORCID IDs are unique identifiers allowing researchers to create a record of their various scholarly contributions in a single place. Please note that ORCID IDs are now *required* for all authors. At resubmission of your final files, please be sure to provide your ORCID ID and those of all co-authors.

Please note that JCB now requires authors to submit Source Data used to generate figures containing gels and Western blots with all revised manuscripts. This Source Data consists of fully uncropped and unprocessed images for each gel/blot displayed in the main and supplemental figures. Since your paper includes cropped gel and/or blot images, please be sure to provide one Source Data file for each figure that contains gels and/or blots along with your revised manuscript files. File names for Source Data figures should be alphanumeric without any spaces or special characters (i.e., SourceDataF#, where F# refers to the associated main figure number or SourceDataFS# for those associated with Supplementary figures). The lanes of the gels/blots should be labeled as they are in the associated figure, the place where cropping was applied should be marked (with a box), and molecular weight/size standards should be labeled wherever possible. Source Data files will be made available to reviewers during evaluation of revised manuscripts and, if your paper is eventually published in JCB, the files will be directly linked to specific figures in the published article.

Journal of Cell Biology now requires a data availability statement for all research article submissions. These statements will be published in the article directly above the Acknowledgments. The statement should address all data underlying the research presented in the manuscript. Please visit the JCB instructions for authors for guidelines and examples of statements at (<https://rupress.org/jcb/pages/editorial-policies#data-availability-statement>).

B. FINAL FILES:

-- Cover images: If you have any striking images related to this story, we would be happy to consider them for inclusion on the journal cover. Submitted images may also be chosen for highlighting on the journal table of contents or JCB homepage carousel.

Images should be uploaded as TIFF or EPS files and must be at least 300 dpi resolution.

****It is JCB policy that if requested, original data images must be made available to the editors. Failure to provide original images upon request will result in unavoidable delays in publication. Please ensure that you have access to all original data images prior to final submission.****

****The license to publish form must be signed before your manuscript can be sent to production. A link to the electronic license to publish form will be sent to the corresponding author only. Please take a moment to check your funder requirements before choosing the appropriate license.****

Thank you for this interesting contribution, we look forward to publishing your paper in Journal of Cell Biology.

Sincerely,

Sandrine Etienne-Manneville, PhD
Monitoring Editor

Andrea L. Marat, PhD
Senior Scientific Editor

Journal of Cell Biology

Reviewer #2 (Comments to the Authors (Required)):

In the revised version of their manuscript that was originally submitted to Review Commons, Kim et al have adequately addressed my previous concerns and provide interesting additional results. The results in the first part (the extracellular environments differentially affect the N-cadherin-mediated migration) and that in the second part (characterization of the leader cells and followers in the collective migration) have been smoothly connected. In addition, the RNA-seq analyses showing little differences (except for the downstream genes of YAP) between the leader cells and followers strongly support their conclusions; the leader and follower cells are exchangeable and YAP is a key regulator for N-cadherin surface levels in these cells. I think the manuscript has greatly been improved and is basically suitable for publication in J Cell Biol, although I noticed only an issue that require further attention:

- N-cadherin differentially regulates the migration of the glioma cell lines in neural cells or ECM, but the mRNA expression levels of integrins, major receptors for ECM, are hardly changed in the cells in these extracellular environments. This is an interesting result, but how N-cadherin inhibits the migration on ECM is unclear. In the Discussion, however, they mentioned the possible involvement of N-cadherin-mediated suppression of focal adhesion formation or activities of Rho family small GTPases, both of which are closely associated with integrin activation. Therefore, it would be better to analyze the activation (or cell surface levels) of the integrin proteins in control or N-cadherin-knockdown glioma cells migrating on the ECM, in addition to the mRNA levels. An antibody for an activated form of beta1-integrin may be useful. Or, they would rewrite the corresponding part of the Discussion.

Reviewer #3 (Comments to the Authors (Required)):

The authors had addressed my concerns, and the revision has improved the manuscript.

RE: JCB Manuscript number: 202401057TR

Corresponding author(s): Jonathan A. Cooper

Reviewer #2 (Comments to the Authors (Required)):

In the revised version of their manuscript that was originally submitted to Review Commons, Kim et al have adequately addressed my previous concerns and provide interesting additional results. The results in the first part (the extracellular environments differentially affect the N-cadherin-mediated migration) and that in the second part (characterization of the leader cells and followers in the collective migration) have been smoothly connected. In addition, the RNA-seq analyses showing little differences (except for the downstream genes of YAP) between the leader cells and followers strongly support their conclusions; the leader and follower cells are exchangeable and YAP is a key regulator for N-cadherin surface levels in these cells. I think the manuscript has greatly been improved and is basically suitable for publication in J Cell Biol, although I noticed only an issue that require further attention:

- N-cadherin differentially regulates the migration of the glioma cell lines in neural cells or ECM, but the mRNA expression levels of integrins, major receptors for ECM, are hardly changed in the cells in these extracellular environments. This is an interesting result, but how N-cadherin inhibits the migration on ECM is unclear. In the Discussion, however, they mentioned the possible involvement of N-cadherin-mediated suppression of focal adhesion formation or activities of Rho family small GTPases, both of which are closely associated with integrin activation. Therefore, it would be better to analyze the activation (or cell surface levels) of the integrin proteins in control or N-cadherin-knockdown glioma cells migrating on the ECM, in addition to the mRNA levels. An antibody for an activated form of beta1-integrin may be useful. Or, they would rewrite the corresponding part of the Discussion.

Paragraph 2 of the Discussion addressed possible ways in which N-cadherin homotypic intercellular interactions could stimulate migration in N-cadherin environments (neurons, astrocytes or N-cadherin-coated surfaces) but inhibit migration in ECM. We favored the hypothesis that leader cell migration is stimulated by N-cadherin engagement on the leading edge but slowed by N-cadherin engagement on the trailing edge. However, we offered alternative explanations, including crosstalk between N-cadherin and integrins or Rho GTPases. At the reviewer's suggestion we have now performed immunofluorescence for focal adhesion proteins including paxillin and total (A11B2) and activated (extended-open conformation, TS2/16) integrin β 1. (The other integrin β chains expressed, β 5 and β 8, do not bind laminin). We did not detect classic focal adhesions in either control or N-cadherin-deficient glioma cells migrating on laminin, even though we used the same antibodies to detect focal adhesions in migrating epithelial cells (Kumar et al., eLife 2023, PMID: 37489578). This result makes it unlikely that crosstalk between N-cadherin and integrins is relevant in this system. Rather than add a supplementary figure showing the essentially negative antibody staining, we have revised paragraph 2 of the Discussion as follows:

“Both the pro-migratory and anti-migratory effects of N-cad on cell migration depended on homotypic N-cad binding to N-cad in the environment or other cells. In addition, α -, β - and p120-catenins inhibited migration in ECM and stimulated migration on neural cells, as expected if their main role is to stabilize N-cad on the surface to increase homotypic binding. A simple hypothesis is that leader cell migration is stimulated by N-cad engagement on the leading edge and slowed by N-cad engagement on the trailing edge (Fig. 3 H). Leading edge stimulation predominates in neural environments but trailing edge inhibition predominates in ECM. This could be explained simply by physical force vectors, but signaling mechanisms may also play a role. For example, N-cad engagement can locally inhibit focal adhesions, and inhibit Rac and activate Rho to repress protrusion and induce cell polarization and contact-inhibition of locomotion (Camand et al., 2012; Julich et al., 2015; Ouyang et al., 2013; Scarpa et al., 2015; Theveneau et al., 2010). **However, crosstalk between N-cad and integrins is unlikely to be relevant in our system because we did not detect classic focal adhesions in either control or N-cad-deficient glioma cells migrating on laminin.** Surface N-cad could alternatively regulate signaling receptors on the same cell, such as stabilizing the fibroblast growth factor receptor as occurs during cortical neuron migration (Kon et al., 2019). Catenins may also regulate directional migration from intracellular vesicles (Vassilev et al., 2017). In our system, however, surface N-cad appears to both stimulate and inhibit migration through homotypic binding or signaling interactions between cells.”